# Embryo Rescue in Plant Breeding

**DOI:** 10.3390/plants12173106

**Published:** 2023-08-29

**Authors:** Ugo Rogo, Marco Fambrini, Claudio Pugliesi

**Affiliations:** Department of Agriculture Food and Environment, University of Pisa, Via del Borghetto 80, 56124 Pisa, Italy; ugo.rogo@phd.unipi.it (U.R.); marco.fambrini@unipi.it (M.F.)

**Keywords:** axenic culture, immature embryo, hybridization, in vitro pollination, plant breeding, post-zygotic barriers, ploidy levels, embryo-lethal mutants

## Abstract

Embryo rescue (ER) techniques are among the oldest and most successful in vitro tissue culture protocols used with plant species. ER refers to a series of methods that promote the development of an immature or lethal embryo into a viable plant. Intraspecific, interspecific, or intergeneric crosses allow the introgression of important alleles of agricultural interest from wild species, such as resistance or tolerance to abiotic and biotic stresses or morphological traits in crops. However, pre-zygotic and post-zygotic reproductive barriers often present challenges in achieving successful hybridization. Pre-zygotic barriers manifest as incompatibility reactions that hinder pollen germination, pollen tube growth, or penetration into the ovule occurring in various tissues, such as the stigma, style, or ovary. To overcome these barriers, several strategies are employed, including cut-style or graft-on-style techniques, the utilization of mixed pollen from distinct species, placenta pollination, and in vitro ovule pollination. On the other hand, post-zygotic barriers act at different tissues and stages ranging from early embryo development to the subsequent growth and reproduction of the offspring. Many crosses among different genera result in embryo abortion due to the failure of endosperm development. In such cases, ER techniques are needed to rescue these hybrids. ER holds great promise for not only facilitating successful crosses but also for obtaining haploids, doubled haploids, and manipulating the ploidy levels for chromosome engineering by monosomic and disomic addition as well substitution lines. Furthermore, ER can be used to shorten the reproductive cycle and for the propagation of rare plants. Additionally, it has been repeatedly used to study the stages of embryonic development, especially in embryo-lethal mutants. The most widely used ER procedure is the culture of immature embryos taken and placed directly on culture media. In certain cases, the in vitro culture of ovule, ovaries or placentas enables the successful development of young embryos from the zygote stage to maturity.

## 1. Introduction

In Angiosperm, the seed is produced by double fertilization. The male gametophyte sperm cell fuses a central diploid cell derived from the union of the two female polar nuclei, creating a triploid endosperm. Simultaneously, the second male sperm cell fuses with the egg cell to produce the diploid zygote. The resulting embryo (2*n*) and endosperm (3*n*) continue to develop within the seed coat derived from the maternal tissues of the egg [1]. In most dicot plants after fertilization, the nucleus of the zygote moves to the apical pole, and the zygote divides asymmetrically to produce a small apical cell that will generate the entire embryo apart from the suspensor and root cap, which are derived from the larger basal cell. The cell division generates a mature embryo organized with an apical–basal polarity and radial concentric tissue layers perpendicular to the axis of symmetry. The apical end is formed by the primary shoot apical meristem (SAM), flanked by one or two cotyledons, while the primary root apical meristem (RAM) is located at the basal end [2]. From the early stages of pattern formation and morphogenesis, the stem cell pools, which are essential for virtually unlimited post-embryonic growth, are placed in the RAM and SAM [3]. Conversely, storage reserves accumulate during the next stage of maturation to prepare the embryo for developmental arrest [4]. In most dicot plants, the endosperm is temporary and, therefore, consumed by the embryo during seed maturation, leaving only a peripheral aleurone-like cell layer next to the seed coat, surrounding the mature embryo [5]. Monocotyledonous zygotes, such as maize, rice, and wheat, also exhibit polarity with the nucleus located at the apical pole. The first transverse division of the zygote produces a two-cell embryo, where the basal cell transforms into a large vesicular cell, while the continuous division of the apical cell gives rise to the quadrant, octant, and dermatogenous stages (proembryos stages) [6]. During the later transitional embryonic stage, which in maize occurs about 7–8 days after pollination (DAP), the adaxial–abaxial axis of the embryo becomes evident. The coleoptile primordium starts protruding from the adaxial region, while the scutellar region develops from the abaxial side. The coleoptile protects the development of the SAM, while the scutellum is analogous to the cotyledon in dicot species. The SAM develops on the adaxial side and produces several embryonic leaves during seed development, while the differentiation of the RAM defines the basal pole of the embryo. In rice, the embryo reaches its mature shape when organ differentiation is complete at 7–8 DAP, although morphogenesis continues as the embryo enlarges. The SAM and RAM are enclosed by the protective coleoptile and coleorhiza, respectively. In grasses, the early vegetative stages of embryonic seedlings are incorporated into the embryo before entering dormancy [7].

Many physical, chemical, and genetic factors influence the development of the zygotic embryo [8]. Disruptions in these factors can result in abnormalities and, in the worst-case scenario, embryo abortion [9,10].

Embryo rescue (ER) is a set of techniques commonly used to rescue immature/mature-lethal embryos, and hybrid embryos generated from interspecific and intergeneric crosses unable to survive in vivo or during traditional plant breeding practices [11,12,13,14,15,16,17,18,19]. The procedure involves excising immature or lethal embryos and culturing them in vitro on a specific nutrient culture medium. The developmental differences between dicots and monocots need to be considered when attempting to recover immature embryos. Nutritional species-specific requirements and the assessment of parameters about growth conditions deserve particular attention to distinguish what happens in the normal embryonic development of dicot and monocot embryos.

During the domestication of plants, many genes controlling resistance to biotic and abiotic stresses were lost. The initial goals of the domestication process were the modification of important traits, such as plant architecture, seed dormancy, fruit and seed size, absence of antinutrients, thorns and waxes, compactness of the ears, absence of seed dispersal, and synchronous fruit ripening. This has led to the phenomenon known as domestication syndrome, where plant survival becomes entirely dependent on human care [20]. Unfortunately, the stress-resistant characteristics of wild ancestor species were often overlooked. However, wild ancestors of crops and/or related wild species constitute a large reservoir of genes that can be introduced into cultivated crops through interspecific and intergeneric hybridization [21]. The rediscovery of Mendelian laws laid the foundation for genetic improvement based on the inheritance of characteristics. However, crossing often requires overcoming physiological and genetic barriers, such as fertilization incompatibility, embryo abortion, and seed dormancy. ER and direct in vitro pollination, such as pollinating stigmas or pistils and opened ovaries or ovules, have proven advantageous in overcoming embryo abortion and pre-zygotic incompatibility barriers, respectively [22]. In vitro pollination techniques have yielded the best results in species with large ovaries containing numerous ovules, including families such as Brassicaceae, Caryophyllaceae, Papaveraceae, Primulaceae, and Solanaceae [23,24]. ER has been instrumental in recovering haploid, dihaploid, doubled haploid (DH), polyploid embryos, addition lines, and substitution lines resulting from interspecific or intergeneric crosses. ER techniques have been also used to shorten the reproductive cycle, propagate rare plants, study the physiology of seed germination, understand embryo nutrition, and study embryo morphogenesis from the zygote to cotyledon stage [12,17,19,25,26,27,28,29,30,31] (Figure 1).

This review provides up-to-date information on ER techniques, their impact, and applications in the genetic improvement in major and minor crops.

## 2. The Origins of Embryo Rescue: A Brief Historical Overview

Charles Bonnet (1720–1793) documented ER for the first time in the 18th century. He achieved remarkable success by excising mature embryos of *Phaseolus vulgaris* and *Fagopyrum* and transplanting them into the soil where they could grow [14]. Later, many scientists started assessing the process with various nutrient media. From 1890 to 1904, ER was evaluated with nutrient media containing salt and sugars along with tissue culture techniques. In 1904, E. Hanning successfully introduced mature embryos into the in vitro tissue culture of some Brassicaceae species (i.e., *Raphanus sativus*, *R. landra*, *R. caudatus*, and *Cochlearia danica*), obtaining seedlings on a saline medium supplemented with sugar. However, he found the problem that in vitro embryo culture originated small and faint seedlings compared to those raised in vivo [32,33]. From these experiments, Hanning focused on the necessity of a high osmotic concentration in the culture medium since the embryos were unable to grow in media lacking sucrose as a carbon source, and the essentiality of an adequate source of nitrogen to allow seedling development from embryos [12,32].

The culture method was used by Brown [34] to assess the efficacy of various organic nitrogenous compounds in promoting the growth of isolated barley (*Hordeum vulgare*) embryos nourished in a mineral saline medium supplemented with sucrose. These experiments demonstrated that amino acids, such as aspartic acid, glutamic acid, and asparagine, served as superior nitrogen sources, leading to increased dry weight and nitrogen content of cultured embryos. Other studies have shown the favorable action of reserve tissues, such as endosperm and cotyledons, in embryo development, although their indispensability was not specifically established [35]. Nevertheless, the reduced growth of *Phaseolus lanatus* embryos deprived of cotyledons and cultured in distilled water solidified with agar was significantly improved by adding glucose or an extract containing reducing sugar from cotyledons of germinated seeds [36]. Andronescu [37] observed the stunted growth of maize (*Zea mays*) seedlings from embryos lacking scutellum, demonstrating the importance of this organ in the absorption of essential nutrients for the development of monocot embryos.

In 1924, Kurt Dieterich was a precursor in testing the possibility of embryo culture both in mature and immature explants [38]. His experiments involved culturing embryos from species of various families, including Solanaceae, Linaceae, Brassicaceae, Polygonaceae, Asteraceae, Cucurbitaceae, and Poaceae, revealing differences in nutrient requirements, such as carbon and nitrogen sources, between immature and mature embryos.

In vivo, embryos derived from interspecific or intergeneric crosses are often underdeveloped and lethal. Frequently, crosses between plants with incompatible genomes lead to early embryo abortion. Often, ER allows the growth of these embryos with reduced viability until development into mature plants. The first application of the in vitro culture of zygotic immature embryos from interspecific cross dates to 1925, when Friedrich Laibach used this technique to prevent embryonic abortion in *Lilium perenne* × *L. austriacum* cross [39]. Laibach’s experiments were fundamental because they gave the initial impetus to the use of interspecific and intergeneric hybridization in genetic improvement, even between species in which the nature of incompatibility did not allow the enormous genetic variability present in wild species to be exploited.

## 3. The Embryo Rescue Techniques

### 3.1. Pre-Zygotic Barriers in Hybridization and Techniques to Overcome Them

Intraspecific, interspecific, and intergeneric hybridizations are important reproductive mechanisms that have contributed to plant speciation and plant breeding [21]. These crosses allow the introgression of advantageous genes from wild species to improve cultivated crops. However, pre- and post-mating reproductive barriers often hinder the application of hybridization techniques in genetic improvement, especially when phylogenetically distant species are involved [40]. Post-mating reproductive barriers can be pre-zygotic and post-zygotic because incompatibility reactions occur before (pre-zygotic) and after (post-zygotic) fertilization, respectively. Notably, the positive fate may depend on the crossing direction for each hybridization.

In the pre-zygotic type, the incompatibility reaction often results in the failure of pollen germination, pollen tube growth, or pollen tube penetration into the ovule, which may occur at various levels in different tissues, such as the stigma, style, or ovary [40,41,42].

To overcome pre-zygotic barriers, cut-style or graft-on-style techniques, application of pollen mixture from several species, placenta pollination, and in vitro ovule pollination have been used [43,44,45]. The techniques to be applied depend on the type of genetic barrier (i.e., incompatibility) that prevents fertilization.

To conduct in vitro fertilization, bringing fully functioning male and female gametophytes into contact and establishing optimal culture conditions to ensure successful gamete fusion is crucial. In pollination methods in vitro, the reproductive organs (stigma and anthers) are isolated, followed by controlled fusion of the male and female gametes. In addition, it is necessary to set up a culture medium that allows the zygote to develop to full maturity, followed by seed germination.

The graft-on-style method is useful when pollen germinates on the stigma but an incompatibility reaction occurs between pollen and style. This type of incompatibility, known as gametophytic incompatibility, causes the pollen tubes to stop and burst after traveling approximately one-third of the length of the style. In this case, the pollination of the entire gynoecium in vitro would not yield fruitful results, as all pollen tubes would be halted at the early part of the style, even under in vitro conditions. To overcome this challenge, the grafting technique is employed, wherein an incompatible style is replaced with a compatible one. This allows the pollen tube to follow its normal growth with the compatible style grafted into the ovary. Alternatively, in species with this type of genetic incompatibility, one proceeds by either shortening the style and micrografting the stigma at the base of the style or by directly depositing pollen on the ovule.

If incompatibility is sporophytic, the recognition substances are deposited in the exine of the pollen during gametogenesis, and the incompatibility reaction occurs at the level of the stigma. Therefore, for the purpose of promoting in vitro fertilization, delivery of pollen on the ovary or ovules may be preferable. Moreover, even placement of germinated pollen grains directly on the decapitated style of the stigma may be equally interesting [27,46].

### 3.2. Post-Zygotic Barriers in Hybridization and Techniques to Overcome Them

When the incompatibility is post-zygotic, the culture of immature embryos may be required. Interspecific and intergeneric hybridizations hinder many post-zygotic barriers [47,48,49], the mechanisms of which are not fully understood yet, and these aspects further complicate the adoption of the best ER techniques. Nevertheless, in vitro culture of immature embryos has emerged as the most widely employed technique, demonstrating remarkable practical success in interspecific and intergeneric hybrids production. These techniques have enabled the transfer of beneficial genes from wild to cultivated species. In higher plants, embryo development is closely linked to the presence of a well-formed endosperm. While the endosperm can develop without the embryo, the absence of endosperm results in early embryo abortion [40,42,50]. Failure to form endosperm commonly accompanies interspecific and intergeneric crosses, and this phenomenon is often present in embryo-lethal mutants. In flowering plants, the endosperm plays a crucial role in coordinating nutrient and hormone supply from maternal tissues to the embryo [51]. Several mechanisms have been proposed to explain hybridization barriers associated with abnormal endosperm differentiation. These include the endosperm equilibrium number (maternal: paternal ratio of 2:1) [52], the difference between the activation and response values expressed by the polar nuclei activation index [53], and genomic imprinting, with the emerging view that it largely influences parental genome dosage [45,54]. Under these circumstances, embryos frequently abort during their development, and ER procedures have proven useful for overcoming these barriers across a wide range of plant species.

Successful results in immature embryo survival depend on the technique adopted (e.g., culture in the ovary, ovule, or embryo). Equally important are the following aspects: the excision procedure, the preservation of embryo integrity, the protocol of sterilization methods, the culture medium composition, and the environmental conditions encompassing light intensity, quality, and temperature. Intrinsic factors, such as embryo size and developmental stage, also play an important role [12,19,25,28,29,55].

In embryo culture, the rescue of a very young embryo is more challenging than an already differentiated one. Ideally, it is more advantageous to recover the immature embryo at a later stage, often referred to as the “autotrophic phase” by some authors [56]. Hence, the embryo is excised from the plant when it reaches the maximum possible development within the constraints imposed by the absence of the endosperm. Typically, excision occurs within the first two weeks after pollination.

Embryo recovery at an early stage of their development, such as the globular or heart stage, may be preferable in certain interspecific combinations. This is because reaching a more advanced stage of development, such as torpedo or cotyledon, often leads to an abortion of the embryo in vivo that is no longer recoverable. However, working with these early embryos presents greater challenges because great is the risk of damaging them during excision, and complex are their nutritional requirements, particularly for proembryos [11].

To overcome these difficulties, the technique of ovule culture allows for the collection of immature embryos within their ovules. This involves in vitro culture of the fertilized egg cell for several days (e.g., one week) followed by the excision of the immature embryo from the cultured ovule. The excised embryo is then transferred to a substrate where it can complete its development [57,58,59]. This method produced new genomic combinations in species, genera, and families of dicot and monocot, for instance, *Nicotiana* [60], *Gossypium* [61], *Brassica* [62,63], *Helianthus* [58], *Tulipa* [64], *Lilium* [65], *Vanilla* [66], *Rhododendron* subgenera or sections [67], and *Vaccinium* [68,69,70], among others.

Generalizing the methods and media used in various in vitro culture techniques is inherently challenging. Each class (dicots or monocots), family, species, and genotype, as well as embryo derived from different hybridizations, requires the setup of specific methodologies to create the most suitable culture medium. This involves several aspects, such as salt concentrations, carbohydrates, vitamins, plant growth regulators (PGRs), and pH, as well as the exploration of proper environmental culture conditions, including temperature, light, and photoperiod. It is most likely that the optimized conditions are species-specific and, therefore, unsuitable for immature embryos of other species or those derived from different crosses. Moreover, dicot species frequently require a multi-step approach, starting from ovule culture to ER, unlike monocot species. This relates to the different types of development that characterize the embryos of dicots compared to monocots. Below, two examples of culturing ovules from phylogenetically distant dicot species are reported.

The in vitro ER technique was employed to obtain embryos from interspecific hybridizations between sunflower and other *Helianthus* species. The embryos were cultured for 7 days in ovules on artificial agarized media [58]. During this culture period, 51 to 84% of embryos of five sunflower genotypes developed to the vascular stage, comparable to greenhouse-grown plants. The survival and development of ovule-grown sunflower embryos were not affected by the salt composition of the four culture media tested: B5 and B5S [71], MS [72], and NN [73], nor did they show any benefit from high sucrose levels (90 or 120 g L^−1^) [58].

The ovule culture technique was utilized to enhance the survival of hybrid embryos between *H. annuus* × *H. maximiliani* Schrad. Approximately, 52% of the hybrid embryos exhibited viability when sunflower ovules were grown on standard media, indicating that the ovules provided the necessary requirements for the early stages of hybrid embryo development. Ovules were isolated and cultured for 14 days on fresh media with the same composition. About 50% of *H. annuus* embryos and 25% of *H. annuus* × *H. maximiliani* hybrid embryos germinated into seedlings. Six *H. annuus* × *H. maximiliani* seedlings were transferred to the greenhouse until flowering. All plants showed intermediate traits inherited from both parents [58].

In another study, Momotaz et al. [59] conducted intergeneric hybridizations to introduce genetic variability in crucifer crops. Seven species of *Brassica* (i.e., *B. campestris* var. *trilocularis*, *B. campestris* var. *pekinensis*, *B. nigra, B. oleracea* var. *capitata*, *B. oleracea* var. *alboglabra, B. juncea* var. *napiformis*, and *B. carinata*) and three species of *Sinapis* (i.e., *S. alba*, *S. arvensis*, and *S. turgida*) were used; reciprocal crosses were also made. In these hybridizations, ER played an essential role in overcoming post-zygotic barriers. Two methods were employed for hybrid plant development: in vitro ovary–ovule culture and direct ovule culture. However, the ovule culture showed better responses in terms of hybrid plant rescued. In this method, young fruits were harvested 15–20 DAP, and ovules were cultured on solidified B5 medium [71] supplemented with 2.5 mg L^−1^ 1-Naphthaleneacetic acid (NAA), 2.5 mg L^−1^ N^6^-Furfuryladenine (kinetin), and 150 mL L^−1^ coconut milk [74]. Both callus formation and direct germination of embryos were observed from ovules. Calli were transferred to an MS medium [72] supplemented with 0.5 mg L^−1^ NAA and 2.5 mg L^−1^ kinetin. Germinated embryos were directly cultured on a hormone-free MS medium. The cultures were maintained at 25 ± 2 °C with a 16 h photoperiod. Seedlings were multiplied in vitro by growing shoots and nodal segments on MS medium, and they were later transplanted in the greenhouse on suitable soil. Specifically, embryos were obtained from the crosses between *B. campestris* var. *trilocularis* × *S. turgida*, *B. campestris* var. *pekinensis* × *S. arvensis*, *S. arvensis* × *B. campestris* var. *pekinensis*, *B. oleracea* var. *alboglabra* × *S. alba*, *B. oleracea* var. *alboglabra × S. turgida*, *B. carinata × S. alba*, *B. carinata × S. arvensis*, and *B. carinata × S. turgida*. The highest efficiency was detected in *B. carinata × S. arvensis* (11.5%) with 29 plants. Reciprocal crosses resulted in no hybrid, except with the combination of *S. arvensis × B. campestris* var. *pekinensis*. Morphological characteristics, chromosome number, and isoenzyme analyses confirmed the hybrid nature of the plants. Among the six combinations, four were identified as true hybrids, one as sesquidiploid, and one as a false hybrid [59].

To overcome the challenges associated with culturing young embryos, an interesting technique provides the relocation of hybrid embryos to a normally developing cellular endosperm dissected from a normal ovule of the parents or a third species, which serves as “nurse tissue”. The embryos and endosperm tissue are then transferred to the surface of the culture medium to activate the in vitro growth of the hybrid embryos [75]. Williams et al. [75] implemented a genetic improvement program involving some types of interspecific crosses, such as *Trifolium ambiguum* × *T. repens*, *T. ambiguum* × *T. hybridum*, *T. repens* × *T. uniflorum*, *Ornithopus sativus* × *O. compressus*, *O. pinnatus* × *O. sativus*, and *Lotus pedunculatus* × *L. corniculatus*. The “nurse tissue” technique was essential for the cross *O. pinnatus* × *O. sativus* hybrids to overcome defective endosperm development and for the cross *T. ambiguum* × *T. hybridum*, which resulted in completely sterile progeny. An interesting feature of the embryo culture on normal endosperm was the high success rate in the apparently normal and complete development of the hybrid embryos. In fact, although at least 100 putative hybrid proembryos were grown per cross, only a few seedlings were transferred into the soil and reached maturity.

The globular embryos of *Capsella bursa-pastoris*, smaller than 50 μm in size, showed higher survival rates when grown in ovules cultured in vitro than when individually inoculated onto the medium [76]. Monnier [77] proposed a solid media culture system for *Capsella bursa-pastoris* proembryos consisting of two concentric rings. The plate contains a central medium suitable to grow very immature embryos and a different peripheral medium more appropriate for the requirements of embryos at a later stage of development.

Elaborate ER methods have also been successfully applied to monocot species. In *Zea mays*, a double-layered medium for the culture of both zygote and proembryos was exploited [78,79]. Briefly, Matthys-Rochon et al. [79] established media for 6 DAP (pre-transitional) and 7 DAP (transitional) maize embryos. Ovaries, containing endosperm and well-organized embryos, were removed and transferred into drops of liquid MS medium [72] supplemented with 0.35 M sucrose. This first stage was indispensable for cell survival. For 7 DAP proembryos, the basal medium was NBM [80] supplemented with different concentrations of sucrose or maltose (0.18 M and 0.26 M) and solidified with gelrite (0.1%, *w*/*v*). The 6 DAP proembryos were first transferred to a bilayer culture system. To support embryo growth, the bottom layer of the N6 basal medium [73,81] was supplemented with 0.09 M sucrose and solidified with 0.8% (*w*/*v*) marine Agarose. The upper layer consisted of N6 medium supplemented with 0.35 M sucrose or 0.30 M maltose, with or without cytokinins, and solidified with 0.8% (*w*/*v*) SeaPlaque Agarose. The best results for transitional embryos (7 DAP) were obtained on media supplemented with 0.25 M maltose, resulting in 86% of plants from 200 cultured embryos. For these embryos, the embryo orientation on the surface of the culture media was crucial. For pre-transitional embryos (6 DAP), the best results (58% of plants from 52 cultured embryos) were obtained on media supplemented with 0.30 M maltose and 0.03 mM of trans-zeatin riboside (ZR). This protocol allowed the production of fertile plants in approximately two months [79].

Embryo and ovule cultures have been employed in tulips to overcome crossing barriers in interspecific hybrids, such as *Tulipa gesneriana* × *T. fosteriana*, *T. gesneriana* × *T. eichleri,* and *T. gesneriana* × *T. greigii* [82] (extensively reviewed by Marasek-Ciolakowska et al. [83]). In intraspecific crosses of *T. gesneriana* × *T. gesneriana*, Custers et al. [84] investigated the optimal conditions for culturing ovules and collecting embryos from pollinated flowers from a few days to 3 weeks. Half-strength MS medium [72] supplemented with 6% sucrose was the most suitable for obtaining larger bulbils. Culturing the ovules in Petri dishes containing a thin layer (6 mm depth) of MS medium [72] resulted in a higher rate of bulbil differentiation compared to cultures in a thick medium. For bulbil formation, the continuous dark condition was superior to a 16 h photoperiod condition. The optimal growing temperature was 12–15 °C. Based on these results, the following method was established to cultivate tulip ovules. The cultures were grown at 15 °C for 12 to 15 weeks, followed by 12 weeks at 5 °C to induce germination. Subsequently, the cultures were grown to 15 °C for 12 to 18 weeks to allow seedling growth and bulbil development. Furthermore, 6 weeks of culture at 15 °C was needed for maturation. Under improved conditions, up to 90% bulbil formation was obtained. Custers et al. [84] also compared embryo and ovule culture methods and showed that ovule culture could be applicable to save small embryos (0.3–0.7 mm in length), while isolated embryo culture yielded satisfactory results with embryos measuring at least 3 mm.

Various ER techniques have been employed to cultivate young embryos, including ovary and placenta cultures. These methods have also been combined, such as ovary culture followed by embryo or ovule culture, as well as placenta culture followed by subsequent culture of the ovary, ovule, and embryo (reviewed by Krishna et al. [19]). Overall, these ER methodologies have since been extensively employed to generate many interspecific and intergeneric hybrids [12,28,29,55,85]. Some examples are the applications of ER in crosses within different genera or families, such as in *Phaseolus* [86], *Solanum* [87], *Cucurbita* [88], Brassicaceae family [63], *Manihot* [89], *Trifolium* [90,91,92], *Vigna* [93], *Elaeis* [94], *Paracurcuma* and *Eucurcuma* subgenera [95], *Alliums* [96], and many other crosses involving phylogenetically distant species (see Section 4). In addition, the enormous potential and diverse applications of ER techniques will be highlighted in next sections, which, with the advent of new biotechnologies (e.g., genome editing), have not been limited to the recovery of lethal embryo hybrids.

### 3.3. Medium Composition and Environmental Factors Suitable for Embryo Rescue Techniques

In vitro culture of plant cells, tissues, and organs requires of a plethora of different media, and therefore, it is impossible to establish a universal culture medium and/or standard environmental conditions suitable for the cultivation of all species and, within a single species, of all genotypes. Various aspects, such as genotype/cultivar, PGRs, chemical composition of the culture medium, and physical factors, influence the development of embryos in vitro. ER, as previously reported, can be divided into several techniques. The formulation of culture media profoundly effects the growth and development of immature and mature embryos, and many formulations have been developed for different plant species, although some are commonly used. The Murashige and Skoog (MS) formulation [72] is recognized as the most used base medium. However, there are other effective media frequently employed, including those reported by White [97], “Gamborg’s B-5” by Gamborg et al. [71], Schenk and Hildebrandt (SH) [98], Nitsch and Nitsch (NN, N6) [73], and “Woody Plant Medium” by Lloyd and McCown [99]. Additionally, plant ER media can include various components and additives, which can be placed into eight categories: water, nutrient salts (micro and macronutrients), vitamins, amino acids, carbohydrates, gelling agents, PGRs, and other organic supplements.

The process of optimizing culture media and environmental conditions is complex and time-consuming, requiring expensive chemicals to set up an efficient and different protocol for each condition.

Given these challenges, applications of artificial neural networks to in vitro plant culture systems could offer valuable insight into the prospects and potential of network technology [100]. These neural networks could be applied to various aspects of in vitro culture, including the composition of the most suitable medium for each proposed culture situation. However, the use of computer-based tools in this context is still limited, despite preliminary work demonstrating the advantages of employing these tools for analyzing large datasets [101,102]. For example, in a study by Hameg et al. [103], an experiment was optimized using three machine learning algorithms: artificial neural networks, fuzzy logic, and genetic algorithms. The objective was to unravel the essential minerals and predict the optimal combination of salts for successful in vitro micropropagation of hardy kiwi (*Actinidia arguta*). The authors demonstrated the suitability of computer-based tools for improving plant in vitro micropropagation of *A. arguta,* predicting a new mineral media formulation that improves growth response and avoids morph-physiological abnormalities. However, this new approach has proven to be valid on a case-by-case basis. Therefore, it should be considered whenever we are faced with different ER situations. In an overall assessment, no chemical/physiological or environmental generalizations can be made about ER, as it remains one of the most complex phases of in vitro culture of immature embryos. Nonetheless, valuable insights can be gained from the excellent reviews [12,19,26,27,28,53] regarding the optimal culture conditions to be used in different species for ER.

## 4. Embryo Rescue Applications

### 4.1. Embryo Rescue in Hybrid Plants Obtained to Introduce Resistance Genes to Abiotic and Biotic Stresses and New Agronomic Traits into Crops

In plant breeding, there is great interest in gene transfer by interspecific or intergeneric hybridization to draw on genetic sources for traits, such as resistance to several stresses, the control of plant architecture, fruit and seed productivity, and the modification of shape and color of flowers, both into dicot and monocot crops. However, crosses within many genera may not occur due to double fertilization failure or early embryo abortion. Hence, hybridization achievement often relies on the support of ER techniques (Figure 1).

#### 4.1.1. Dicot Species

Extensive hybridization programs have been implemented in the genus *Helianthus* among cultivated sunflower (*Helianthus annuus*) and wild species to introgress genes conferring resistance to biotic and abiotic stresses [104,105,106]. Notably, new interspecific hybrids, such as *H. annuus* × *H. hirsutus*, and *H. annuus* × *H. scaberimus*, have been produced [107]. Remarkably, hybrids have been produced using *H. annuus* as a female parent, crossing it with *H. maximiliani* [108], *H. nuttalii*, *H. mollis*, and *H. strumosus* [109]. In all these crosses, ER was indispensable for the survival of the hybrid embryos. For example, Krauter et al. [108] made 33 different hybrid combinations using *H. annuus* as both female and male parents. Fertilized seeds (5–10 DAP) of the different crosses were collected. After sterilization, immature embryos were isolated. Embryos were cultured on different media according to their stage of development. For small embryos (0.2–1.5 mm), a modified B5 medium [71] supplemented with 90 g L^−1^ sucrose was used. Embryos over 1.5 mm were grown on a modified MS medium [71,108] supplemented with 10 g L^−1^ sucrose. Embryos were maintained at 25 °C under continuous light. As soon as the small embryos reached a size of 2–3 mm, they were grown in MS medium. Well-formed shoots were transferred to the greenhouse, where they developed into vigorous plants. In the genus *Helianthus*, this ER protocol showed good efficiency. In fact, 481 hybrid plants (41%) were recovered from 1178 isolated embryos.

In *Nicotiana*, hybrid lethality has been observed in some combinations of interspecific crosses. *N. tabacum* (2*n* = 4*x* = 48, genome SSTT) has an S subgenome derived from *N. sylvestris* (2*n* = 2*x* = 24) and a T subgenome derived from *N. tomentosiformis* (2*n* = 2*x* = 24), the progenitors of tobacco. Hybrid seedlings from the *N. stocktonii* × *N. tabacum* cross have been successfully rescued through ovule culture [110], as they did not highlight lethality. The flowers of *N. stocktonii*, used as the female parent, were emasculated and fertilized with *N. tabacum* pollen. At 6, 9, and 12 DAP, *N. stocktonii* fertilized ovules were taken. The ovules were cultured on half-strength MS medium [72] supplemented with 3% sucrose at 28 °C under continuous illumination. Viable hybrid plants were obtained only from 6 DAP ovules (6 viable plants/525 cultured ovules). Conversely, hybrid seedlings from the *N. stocktonii × N. sylvestris* cross were lethal. Moreover, hybrid seedlings from *N. stocktonii* × *N. tomentosiformis* cross also manifested lethality. These results suggested that both S and T genomes harbor factors causing hybrid lethality in *N. stocktonii*. Muraida and Marubashi [110] proposed that in *N. tabacum*, the factors responsible for lethality must have been lost due to the reorganization and modification of the genome during and after amphidiploid formation.

In the genus *Nicotiana*, ER techniques have also been used to support interspecific hybridization between *N. glutinosa* (2*n* = 24) and *N. megalosiphon* (2*n* = 40). *N. glutinosa* was used as the female parent in the cross with *N. megalosiphon*. Immature ovules at 7 DAP were grown on NN medium [73]. In addition, the MS medium [72] supplemented with 6 mg L^−1^ NAA and 2 mg L^−1^ 6-Benzylaminopurine (BAP) was used for callus induction. Shoots were regenerated on MS supplemented with 2 mg L^−1^ indole-3-acetic acid (IAA) and 2 mg L^−1^ kinetin. For root induction, MS medium was supplemented with 3–5 mg L^−1^ IAA and 2 mg L^−1^ kinetin. About 50 seedlings were obtained and grown to maturity. The hybrids had a wide range of phenotypic variations (e.g., plant height, leaf shape, flower color, and corolla size), probably attributable to the different number of chromosomes detected in the somatic cells of the plants (2*n*, from 28 to 32) [111].

In chrysanthemum, ER techniques have been employed to develop interspecific hybrids for aphid resistance [112], heat tolerance [113], drought tolerance [114], cold tolerance [15], salinity tolerance [115], and heterosis [116] (Table 1).

For instance, Zhu et al. [115], to obtain salinity tolerance, generated an intergeneric hybrid between *Chrysanthemum* × *morifolium* (2*n* = 6*x* = 54) and *Artemisia japonica* (2*n* = 4*x* = 36). For the ER, ovaries were removed at 12–15 DAP and sterilized. Ovules were excised and transferred to MS medium [72] containing 2 mg L^−1^ of BA and 0.5 mg L^−1^ of NAA. Cultures were kept at 25 °C with a 16 h photoperiod. Seedlings were kept on the culture medium until complete rooting. Rooted seedlings were kept in a 16 h photoperiod at 19 ± 1 °C in half-strength MS medium for 1 week. The plants were acclimatized before transfer to soil. The hybrid plants manifested morphological characteristics intermediate to the parents. In particular, the hybrids possessed high chlorophyll and proline contents and lower NA^+^ ion concentrations, making them more tolerant to salt stress [115].

In bluebells, new flower shape and cold tolerance have been developed in intraspecific *Campanula carpatica* and interspecific (*C. medium × C. formanekiana*) hybrids [117]. However, a frequent problem encountered in these crosses is defective endosperm development, which does not allow the hybrid embryos to complete the maturation in planta. To overcome this problem, ovules from 7 to 30 DAP were isolated and cultured on a modified MS medium [72]. In culturing the ovules, those taken between 14 and 30 DAP provided better embryo germination than ovules taken at 7 DAP. Actually, ovaries were also cultured in vitro in the same experiment. The germination rate of hybrid embryos from ovaries was higher than that of isolated ovules.

Interspecific hybrids between 2 cultivated gentian species (*Gentiana scabra* and *G. triflora*) and 11 wild species were obtained to broaden genetic variation in breeding programs (Table 1). Ovule culture in vitro played a crucial role in the proper development of hybrid embryos, and the hybrid nature of the resulting plants was confirmed using morphological traits and molecular markers [118]. Ovules were cultured on solidified half-strength MS medium [72] supplemented with 3% sucrose and 1 mg L^−1^ Gibberellic acid (GA_3_) at 20 °C under a 16 h photoperiod. Well-developed seedlings were transferred to the same medium supplemented with 3% sucrose under the same environmental conditions. Rooted plants were grown in pots until flowering. When *G. scabra* was used as the female parent in crosses, interspecific combinations significantly affected seedling yield, which ranged from 0.3 to 427.7 normal seedlings per flower. In crosses of *G. triflora* with five wild species, normal seedling yields ranged from 0.4 to 228.3 per flower. In this instance, significant differences between genotypes were found between both interspecific combinations and reciprocal crosses [118].

Ovule culture was used for immature embryos obtained from an interspecific cross between *Camellia chrysantha* and *C. japonica*, with *C. chrysantha* as the seed parent, to improve genetic variability [119]. Immature embryos were harvested at 90 to 130 DAP and cultured on a modified MS basal medium [72]. The highest number of differentiated adventitious embryos were obtained when ovules were recovered at 110–120 DAP. Better results were obtained by adding polyvinylpyrrolidone (PVP) to the culture medium and adjusting the pH to 6.5. All plantlets proved to be interspecific hybrids using HPLC analysis of leaf polyphenols and random amplified polymorphic DNA (RAPD) analysis [119]. To transfer the yellow corolla color of *C. chrysantha* to *C. japonica*, Nishimoto et al. [120] obtained hybrids with a pale-yellow color through the cross *C. chrysantha* × *C. japonica*. However, subsequent backcrosses of the hybrid with *C. chrysanta* resulted in progenies with deep-yellow flowers (Table 1).

Interspecific hybrids between *Gypsophila paniculata* and *G. manginii* are incompatible with ordinary cross-breeding methods. Kishi et al. [121] obtained, through ovule–embryo cultures, useful hybrid seedlings that reached maturity. Morphological traits and chromosome counting confirmed the hybrid nature of the plants. These hybrid plants exhibited double flowers with a color of pale purplish pink, a valuable character for the flower market [121].

Very important crops are part of the Brassicaceae family. In the genus *Brassica*, there are three diploid species: *Brassica rapa* (2*n* = 2*x* = 20, genome AA), *Brassica nigra* (2*n* = 2*x* = 16, genome BB), and *Brassica oleracea* (2*n* = 2*x* = 18, genome CC), and three related allotetraploids: *Brassica juncea* (2*n* = 4*x* = 36, genome AABB), *Brassica napus* (2*n* = 4*x* = 38, genome AACC), and *Brassica carinata* (2*n* = 4*x* = 34, genome BBCC) [122]. Among these species, hybridization can be natural or artificially induced. However, in some hybrid combinations, especially when the parents belong to different genera carrying genes important for genetic improvement, embryos have difficulty developing normally. Therefore, in Brassicaceae, numerous novel F_1_ hybrids have been produced through the use of various ER techniques: embryo culture [123,124], ovary culture [125,126,127,128], ovule culture [59], and placenta culture [129]. For example, high numbers of F_1_ hybrid plants have been generated from interspecific and intergeneric hybridizations between *Brassica* crops and wild relatives in the genera *Brassica*, *Sinapis*, *Diplotaxis*, *Moricandia*, *Eruca*, and *Orychophragmus* using ER techniques to develop improved crop cultivars [63,130]. In particular, synthetic amphidiploid lines, alien gene introgression lines, alloplasmic lines, monosomic alien chromosome addition lines, and monosomic alien chromosome substitution lines have been produced [63]. Attempts were also made to obtain intergeneric hybrids from the crosses of *B. campestris*, *B. nigra*, *B. oleracea*, *B. juncea*, *B. napus,* and *B. carinata* with *Enarthrocarpus lyratus*. Hybrids using *E. lyratus* as a female parent were obtained through ovary and/or ovule culture in four combinations: *E. lyratus* × *B. campestris*, *E. lyratus* × *B. oleracea*, *E. lyratus* × *B. napus*, and *E. lyratus* × *B. carinata* [131], aiming to introduce genes conferring fungal pathogen resistance from wild relatives to crop species. ER was performed from pollinated ovaries at 4–6 DAP. The ovaries were cultured in MS medium [72] supplemented with 400 mg L^−1^ casein hydrolysate. Ovule culture was conducted at a more advanced stage from pollination. Ovules were excised from 10–15 DAP pollinated ovaries and grown in the same medium. The seedlings obtained from the ovules were used for plantlet multiplication. Shoot meristems were multiplied on MS medium containing 0.5 mg L^−1^ BAP. Roots were induced on MS medium supplemented with 0.1 mg L^−1^ NAA. Seedlings were transferred to soil and grown to maturity. Reciprocal crosses showed strong pre-fertilization barriers and did not produce hybrids except in one combination, *B. juncea* × *E. lyratus* [131].

In *Brassica napus*, the Polima (*pol*) CMS (cytoplasmic male sterility) is the most universally valued male sterility form [132]. It is utilized in hybrid rapeseed production along with the fertility restorer gene *Rfp* [132]. Formanová et al. [133] successfully transferred the *pol* cytoplasm and *Rfp* from the amphidiploids *B. napus* to the diploid species *B. rapa*. They generated a DH *pol* cytoplasm in a *B. rapa* population that segregates for the *Rfp* gene. This achievement involved interspecific crosses, in vitro rescue of hybrid embryos, backcrosses, and a microspore culture [133]. The use of DHs should help in *Rfp* gene mapping (see also Section 4.2.1). For ER, siliques were taken after 30 DAP. Embryos manifested extremely stunted development and were removed and cultured on half-strength MS medium [72] with or without 0.2 mg L^−1^ of IAA. The germinated seedlings were transferred to soil until maturity.

Auxin herbicide resistance from *Brassica kaber* was transferred to *Brassica juncea* and *Brassica rapa* by traditional selection in combination with in vitro ER [134]. Immature siliques were collected from 3 to 5 DAP. After sterilization, the siliques were grown in two different culture media: a medium consisting of MS salts [72] and Gamborg’s vitamins [71] supplemented with 3% sucrose and 500 mg L^−1^ casein hydrolysate, and a medium consisting of MS salts [72] and Gamborg’s vitamins [71] supplemented with 3% sucrose, 0.5 mg L^−1^ NAA, and 2.5 mg L^−1^ kinetin. The siliques were cultured on these media for two weeks. Subsequently, the ovules excised from the siliques were cultured on the same media described above. All cultures were incubated at 24 °C under a 16 h photoperiod. After about one month, the ovules regenerated hybrid seedlings. The seedlings were transferred to a medium containing MS salt [72] and Gamborg vitamins [71] supplemented with 1.5% sucrose for root development. Hybrid plants were cloned by culturing nodal segments. Nodal segments with well-developed roots and shoots were transferred to the soil. Six hybrid plants were obtained from the *B. juncea × B. kaber* (*Sinapis arvensis*) cross from 40 cultured immature embryos. Differently, a higher number of hybrid plants was obtained from the cross *B. rapa × B. kaber*. In fact, 32 hybrid plants were obtained from 50 recovered immature embryos. The fertility and ability to produce backcross progeny of the hybrid plants were also tested, and the auxinic-herbicide-resistant trait was introgressed into *B. juncea* through backcross breeding [134].

A similar procedure was used to introduce tolerance to dicamba (an auxin herbicide) from wild mustard (*S. arvensis*) into canola. The ER technique followed by conventional breeding was used [135]. Immature siliques were collected from 3 to 5 DAP. The siliques were cultured for two weeks on two different culture media: an MS [72] with Gamborg vitamin [71] medium supplemented with 3% sucrose and 500 mg L^−1^ casein hydrolysate, and a medium containing MS salts [72] with Gamborg vitamins [71], 3% sucrose, 0.5 mg L^−1^ NAA, and 2.5 mg L^−1^ of kinetin. Subsequently, the ovules recovered from the siliques were cultured on the same medium. All cultures were incubated at 24 °C with a 16 h photoperiod. The regenerated seedlings were transferred into a medium containing MS salts [72] with Gamborg vitamins [71] and 1.5% sucrose. Nodal segments, cultured in the same growing medium, were used to multiply well-rooted seedlings. Rooted nodal segments were transferred into the soil. Intergeneric hybrids between *S*. *arvensis* (2*n* = 2*x* = 18) and *B*. *napus* (2*n* = 4*x* = 38) were produced, and dicamba tolerance was introgressed into *B*. *napus* through seven generations of backcrossing.

Pen et al. [136] performed a comprehensive series of interspecific crosses to introduce desirable characteristics such as orange/yellow color in cabbage and increased anthocyanin biosynthesis in Chinese cabbage (*B. rapa* subsp. *pekinensis*). Interspecific hybridizations between Chinese cabbage and cabbage were conducted using Chinese cabbage cultivars as the female parent. Additionally, Chinese cabbage cultivars have been used as male parents in crosses with allopolyploids, *B. juncea* and *B. napus*. The use of ER was crucial for the successful development of hybrid embryos. Isolated ovules were cultured on solidified MS medium [72] with 30 g L^−1^ sucrose. The cultured embryos were maintained at 23 ± 2 °C under a 13 h photoperiod, and germinated embryos produced shoots within 15–20 days after isolation. Rooted plants were obtained from shoots grown on MS basal salts within 35–40 days. The authors showed that embryos at 15 DAP were the most suitable material for ER in *B. rapa × B. oleracea* crosses, while embryos at 20 DAP were ideal for allopolyploid–diploid ((*B. napus × B. juncea*) *× B. rapa*)) crosses. Through their breeding program based on wide interspecific hybridization and selection, Pen et al. [136] successfully selected cabbage plants with orange/yellow inner leaves and Chinese cabbage leaves with higher anthocyanin content. These results hold the potential to produce interspecific hybrids and the development of biofortified cultivars within the genus *Brassica* (Table 1).

Ton et al. [137] reviewed the traits that should be molecularly characterized and mapped. This would make their transfer into canola and other Brassicaceae species more feasible. The authors reported both genes for resistance to biotic (e.g., blackleg disease, caused by *Leptosphaeria maculans*; clubroot, caused by *Plasmodiophora brassicae*; and stem rot, caused by *Sclerotinia sclerotiorum*) and abiotic (e.g., heat, drought, cold, and salinity) stresses. They highlighted a study on pan-genomics of *B. oleracea*, which identified many candidate genes for disease resistance in wild species *B. macrocarpa* [138]. Recently, the clubroot-resistant gene *CRd* from Chinese cabbage was transferred to canola through interspecific hybridization [139]. Powdery mildew, caused by *Erysiphe cruciferarum*, is another epidemic pathogen of rapeseed. Gong et al. [140] found a cultivar of *Brassica carinata* with powdery mildew resistance. Through interspecific hybridization followed by a molecular-marker-assisted selection (MAS) program, pathogen resistance was introduced into *B. napus* (Table 1). The use of ER-assisted interspecific hybridization, the MAS program and genome editing were suggested as suitable approaches for improving canola’s resistance to both biotic and abiotic stresses.

Some varieties of *Brassica nigra* are resistant to blackleg disease and have been used in a breeding program to introduce resistance to *Leptosphaeria maculans* in *B. rapa* [141]. Triploid hybrids (2*n* = 3*x* = 27, genome ABC) were produced through the cross of *B. napus × B. nigra* with the support of ER, and allohexaploid hybrids (2*n* = 6*x* = 54, genome AABBCC) were obtained by a chromosomal doubling of triploids. These hybrids demonstrated resistance to the disease inherited from the parent *B. nigra.* Therefore, cytological analysis and in situ hybridization suggested the introgression of blackleg resistance from the B genome of *B. nigra* into the genomes of *B. rapa*. However, allohexaploids displayed extreme genomic instability [141].

Three clubroot-resistant Chinese cabbage varieties were identified and used in an ER-supported hybridization program to transfer clubroot resistance genes to *B. napus* [142]. The ER technique involved the collection of embryos at 15 DAP and culturing them in liquid B5 medium [71] supplemented with 2% sucrose. For the early stages of development, the cultures were placed at 25 °C under a 16 h photoperiod. Once the cotyledons were visible, embryos were transferred to a solid B5 medium [71] supplemented with 2% sucrose. Cytological analysis and MAS were used to identify 297 true hybrids, and among them, 159 hybrids were detected as clubroot resistant using molecular markers linked to resistance genes. These varieties were introduced in improvement programs for introgression resistance in commercial lines of *B. napus* [142] (Table 1).

Cauliflower (*Brassica oleracea botrytis* group, 2*n* = 18, genome CC) is one of the most extensively grown *Brassica* vegetables worldwide. However, it is susceptible to black rot disease caused by the bacterium *Xanthomonas campestris* pv. *campestris* (Xcc) (Pammel) Dowson, which is a major pathogen of the *Brassica* genus [143]. *Brassica carinata* A. Braun (2*n* = 4*x* = 34, genome BBCC) is an important oilseed crop native to the Ethiopian plateau. *B. carinata* possesses genes controlling resistance/tolerance traits for both biotic and abiotic stresses, including resistance to Xcc [144,145]. Hybridization and backcrossing were performed to transfer the black rot resistance genes present in the B genome of *B. carinata* to cauliflower, resulting in the generation of interspecific hybrids and backcross progeny (BC_1_) [146]. However, hybridization between diploid and tetraploid species is challenging due to various reproductive barriers occurring at different stages, such as pollination incompatibility to pre- and post-fertilization barriers [146]. Overcoming these barriers using ER has proven effective in developing interspecific hybrids in *Brassica* [59,147,148]. Successful application of the direct in vitro ovule culture technique facilitated the development of mature hybrid plants (i.e., *B. oleracea* × *B. carinata*), and marker analyses confirmed the introgression of black rot resistance into the interspecific BC_1_ population [146]. Several attempts were also made to transfer some genes from alien *Brassica* spp. To *B. oleracea*, such as resistance to powdery mildew [149], downy mildew [150], black rot [151], atrazine [147], and male sterility [152] (Table 1).

The diversification of CMS is essential to mitigate the risk of pathogen attacks. Bathia et al. [153] developed a novel CSM system by creating sesquidiploids and backcross progenies involving the cybrid *Brassica napus* with the cytoplasm of *B. tournefortii* (*Tour*) and *B. oleracea*. In vitro culture of ovaries and ovules was essential to recover interspecific hybrids. The pollinated buds of the male sterile line were cultured in media at five different intervals from pollination. For example, in the cross between *B. napus* × PSBK-1, 372 ovules taken at different stages of development were cultured. None of the ovules were vital when sampled at 4, 13, and 16 DAP. In contrast, 14 mature ovules were obtained from isolation at 7 and 10 DAP. These results demonstrate how critical the timing is for efficient ovule culture. The backcross population developed with the *Tour* cytoplasm will be helpful for the development of a stable CMS system in *B. oleracea* as an alternative to the *Ogura* CSM (Table 1).

Interspecific hybrids of *Brassica juncea × B. tournefortii* were obtained through the in vitro ER of hybrid embryos, as demonstrated by Kumar et al. [154]. Embryos were harvested from ovules at 20, 25, and 30 DAP and cultured on MS basal medium [72] with three different combinations of PGRs [154]. Cultures were grown at 25 °C under a 16 h photoperiod. The regenerated shoots were transferred to MS medium supplemented with 2 mg L^−1^ of BAP and 0.2 mg L^−1^ of NAA. Plants were transferred to soil and grown to maturity. Notably, a hybrid plant exhibited tolerance to aphid attacks, and it was partially fertile. In addition, the F_2_ and BC_1_ progenies displayed a wide range of morphological traits, and several plants with desirable characteristics, such as reduced aphid infections, drought tolerance, and high yield, were selected [154] (Table 1).

In the *Cucumis* genus, strong sexual incompatibility in interspecific and intergeneric crosses restricted the rapid development of new cultivars with desirable traits, such as disease resistance, insect resistance, flavor, and sweetness [155,156]. It would be advantageous to introduce new genes to increase genetic variability and improve the quality and productivity of melon fruits. Over the last 25 years, various biotechnological techniques have been developed. In addition to genetic transformation and in vitro regeneration to exploit somaclonal variability in embryos and adventitious shoots, ER has proven useful for the introgression of traits from wild species [156]. For example, hybrid embryos were successfully recovered through ER after interspecific crosses, such as *C. melo × C. metuliferus* [157], *C. metuliferus × C. anguria* [158], and *C. melo × C. anguria* L. var. *longipes* [159].

Interspecific hybridization in the genus *Trifolium* using conventional crossing techniques was largely unsuccessful. Post-zygotic barriers are the primary cause of reproductive isolation, resulting in endosperm disintegration and abnormal differentiation, leading to the starvation of the hybrid embryo. Consequently, the application of ER techniques has been employed to save embryos from new combinations of interspecific hybrids (reviewed by Roy et al. [92]). ER has been utilized in interspecific hybridization programs involving *Trifolium ambiguum*, *T. pratense*, *T. montanum*, *T. occidentale*, *T. isthomocarpum*, *T. repens*, *T. nigrescens*, *T. uniflorum*, *T. sarosiense*, *T. alexandrinum*, *T. apertum*, *T. resupinatum*, *T. constantinopolitanum*, *T. rubens*, and *T. alpestre* [92]. For example, the ER technique was successfully used in the cross *T. alexandrinum × T. constantinopolitanum*. Pollination was carried out 2 days after emasculation. Embryos at the heart stage were grown on basal MS medium [72] supplemented with 2.3 μM kinetin and 3% sucrose. From more than 600 crosses, 33 embryos were taken and cultured. Shoots emerged between one and two weeks after inoculation. Seedlings survived only after inoculation with *Rhizobium*. The hybrid plants showed intermediate morphological characteristics, with 55 to 65% pollen fertility and a chromosome number of 2*n* = 2*x* =16.

Interspecific hybridization has played a crucial role in the genetic enhancement in yield in chickpeas (*Cicer arietinum* L.). Wild species and old cultivars provide a wider genetic base for crop species and provide a potential source of resistance genes against biotic and abiotic stresses, in particular, tolerance to cold and drought conditions, as well resistance to *Fusarium oxysporum* f. sp. *ciceris*, cyst nematode (*Heterodera ciceri*), *Ascochyta rabiei*, phytophthora root rot, *Botrytis cinerea*, leaf miner and bruchids, and pod borer (*Helicoverpa armigera*) [160,161,162,163,164]. Among the different annual wild *Cicer* species, the successful hybridization of *C. arietinum* has been achieved only with *C. reticulatum* and *C. echinospermum* [162,163]. Conversely, significant pre-fertilization barriers were observed in the *C. arietinum × C. songaricum* cross, including poor pollen grain germination and a lack of pollen tube penetration in the style tissues [165]. Moreover, strong post-zygotic reproductive barriers in the cross *C. arietinum × C. pinnatifidum* resulted in poor seed set, high hybrid embryo necrosis, albinism, and poor aptitude to form roots [166]. The occurrence of embryo abortion in all crosses, except with *C. reticulatum*, clearly demonstrated the presence of post-fertilization barriers to reduce cross ability. Excising fertilized ovules and cultivating them on artificial media could offer a potential means to overcome these barriers and facilitate the transfer of desirable traits from these species [167,168,169,170].

In many regions of the world, the production of beans faces significant challenges due to various biotic and abiotic stresses, such as *Ascochyta* blight, bean golden mosaic virus (BGMV), and bean fly. To combat these diseases, interbreeding with *P. coccineus* L. or *P. polyanthus* Greenm. has been found to enhance resistance in *P. vulgaris* [171]. The authors showed that the use of *P. coccineus* and *P. polyanthus* as female parents was essential for successful interspecific crosses, which prevent rapid reversal to the recurrent parent *P. vulgaris*. However, post-zygotic incompatibility barriers cause an early hybrid embryo abortion, limiting the success of F_1_ crosses. Therefore, Geerts et al. [171] optimized an ER technique to improve hybridization and produce healthy hybrid plants. In this method, pods at 2 DAP were recovered. In aseptic conditions, the immature embryos were isolated and cultured. Then, dehydrated embryos were grown in a culture medium for germination. Plantlets were obtained after rooting. Subsequently, healthy plants were acclimated by raising them in the soil until maturity.

*Capsicum baccatum* has been reported as a valuable source of genetic variation for many different traits to improve common pepper (*C. annuum*). However, the presence of strong barriers to interspecific hybridization has hindered successful crosses. To overcome these barriers, Manzur et al. [172] performed a comparative study involving two hybridization approaches: (i) genetic bridge species using *C. chinense* and *C. frutescens*, and (ii) direct cross between *C. annuum* and *C. baccatum* combined with in vitro ER. The results showed that *C. chinense* served as an effective bridge species between *C. annuum* and *C. baccatum*, while *C. frutescens* gave poor results. Concerning the ER strategy, the best success was achieved with the *C. annuum* (female) × *C. baccatum* (male) cross. Furthermore, the production of plants from the first backcrosses to *C. annuum* (BC_1_) was only possible through the in vitro culture of immature embryos [172]. In particular, Manzur et al. [172] performed about 2500 hybridizations. Then, immature fruits of about 25–30 DAP were collected. A solidified half-strength MS medium [72] was supplemented with 0.01 mg L^−1^ IAA, 0.01 mg L^−1^ GA_3_, 0.01 mg L^−1^ zeatin, and 40 g L^−1^ sucrose. Embryos were cultured at 25 ± 1 °C for first five days, in complete darkness. After day 6, cultures were incubated under a 16 h photoperiod. In total, more than 1000 immature seeds were cultured. After 30 days of culture, healthy seedlings were acclimated in vivo. F_1_ seedlings at the four-leaf stage were transferred to pots in the greenhouse for morphological and hybridity evaluation by analysis of simple sequence repeats (SSR) marker. As reported previously, using *C. baccatum* as the male parent provided the highest efficiency in fruit set and the number of regenerated hybrids. Fruit set was achieved in 67% of *C. annuum × C. baccatum* combinations, although the efficiency rate varied greatly (from 4 to 30%) depending on the genotypes used in the cross.

Tomato (*Solanum lycopersicum* L.) has been hybridized with the sexually incompatible *L. peruvianum.* However the endosperm failed to develop, resulting in embryo abortion. The hybrid embryo was carefully isolated on a tissue culture medium, and typical development occurred [173]. This approach has proven effective in transferring some resistance to various pathogens in cultivated species, including tobacco mosaic virus, tomato spotted wilt virus, root-knot nematode, and pathogenic fungi.

Resistance to papaya ringspot virus has been transferred to papaya from the incompatible and resistant *Carica cauliflora* using ER to facilitate the development of hybrid embryos [174]. The fruits were harvested at 65 DAP and sterilized. The ovules were excised and cultured on solidified and modified MS medium [72]. The amount of iron was reduced, and the medium was supplemented with 60 g L^−1^ sucrose, 400 mg L^−1^ glutamine, and coconut water (20% *v*/*v*). The callus produced from the ovules was removed and subcultured on the same medium. Subsequently, the callus was transferred into a liquid medium to induce efficient somatic embryogenesis. Well-differentiated regenerated seedlings were obtained from somatic embryos cultured on a medium supplemented with BA and NAA. All cultures were maintained at 25 °C under a 16 h photoperiod. Sixty seedlings derived from somatic embryos were subjected to isoenzyme analysis. The results suggested that the somatic embryos obtained from the ovules of interspecific hybridization between *C. papaya* and *C. cauliflora* were probably of hybrid origin. In the hybrids, the undesirable wild traits were removed from the breeding population by performing successive backcrosses with cultivated papaya [174].

Wild soybean species have been intensively used to extend the genetic variability of *Glycine max* in breeding programs (reviewed by Chung and Singh [175]). In these crosses, ER ensured key support in obtaining healthy hybrid plants [176,177,178,179,180,181]. Remarkably, wild perennial *Glycine* species harbor several resistance genes toward soybean rust (*Phakopsora pachyrhizi* Sydow) [182,183,184], soybean cyst nematode (*Heterodera glycines* Ichinohe) [185], soybean brown spot (*Septoria glycines* Hemmi.) [186], alfalfa mosaic virus [187], bean pod mottle virus [188], white mold *(Sclerotinia sclerotiorum* (Lib.) de Bary) [189], 2,4-Dichlorophenoxyacetic acid (2,4-D) [190], and chloride [191]. With the development of a wide cross methodology, these valuable traits might be successfully transferred to the soybean. For example, Bodanese-Zanettini et al. [177] obtained successful crosses using 9 Brazilian soybean lines as female parents and 12 accessions belonging to *Glycine canescens*, *G. microphylla*, *G. tabacina*, and *G. tomentella*. Immature hybrid pods were harvested between 20 and 30 DAP. After pod sterilization, embryos were excised and cultured. The cultures were maintained at 25 ± 1 °C with a 16 h photoperiod. Embryos were cultured in B5 medium [71] supplemented with 500 mg L^−1^ glutamine, 100 mg L^−1^ serine, 250 mg L^−1^ asparagine, 250 mg L^−1^ casein hydrolysate, 1 μM BAP, 0.1 μM NAA, and 4% sucrose. Embryos were excised at different stages of development. Therefore, after about 20 days (for larger embryos) or 60 days (for smaller embryos), the embryos were transferred to B5 medium [71] with 10% sucrose and 0.5% activated charcoal. After two months, mature and dormant embryos were transferred to SH medium [98] supplemented with 1% sucrose. Rooted seedlings were transplanted into pots and brought to maturity. About 37% of the immature embryos developed into seedlings, and about 8% produced mature plants. Cytological chromosome number analyses conducted on root apices and at spore meiosis, combined with leaf isoenzyme analyses, were used to identify the hybrid status of the obtained plants [177]. More recently, Singh and Nelson [192] optimized methods to rescue F_1_, amphidiploid, BC_1_, BC_2_, BC_3_, and fertile soybean plants derived from crosses of soybean with 26 wild perennial *Glycine* species (e.g., *Glycine tomentella*, *G. argyrea,* and *G. latifolia*). The development of an efficient ER method was essential to achieve these results. Although most F_1_ seeds failed to reach maturity and were aborted, seeds from 19 to 21 DAP pods were cultured aseptically in various medium formulations, leading to the development of hybrid plantlets [192]. Morphological variants among 40-chromosome plants were identified in both BC_3_F_2_ and BC_4_F_2_ generations. These characteristics include, among others, white flowers, delayed flowering, seeds shattered after pod maturation, plant vigor, broad leaves, seed coat color, and soybean rust resistance [192].

Downy mildew, caused by *Plasmopara viticola* (Berk. and Curt.) Berl. and de Toni, is a destructive grapevine disease commonly controlled worldwide through fungicide treatments [193]. However, conventional breeding approaches also rely on natural sources of resistance found in wild grapevine (*Vitis*) species. A highly efficient ER technique was developed to support the survival of hybrid embryos obtained from crosses using *Vitis vinifera* cultivars as female parents and Chinese wild *Vitis* species as male parents. The aim was to breed disease-resistant and seedless grape cultivars [194]. More recently, new high-quality cultivars of seedless and disease-resistant grapes were obtained from a stenospermocarpic *Vitis vinifera* L. used as female parents in a cross with Chinese wild *Vitis* [195]. ER was used to rescue immature hybrid embryos to prevent their degradation, and several F_1_ hybrids were successfully obtained. Molecular markers were used to identify more than 300 seedless plantlets, and 2 F_1_ progenies resistant to downy mildew were detected [195].

Giancaspro et al. [196] developed a complex yet effective ER method for obtaining seedless fruits in table grape varieties. After crosses, immature fertilized ovules from stenospermocarpic berries were grown on culture media suitable for ovule and embryo development. The ovules were maintained for 8 weeks at 25 °C in the dark, with regular transfers to fresh medium every two weeks. The ovule culture medium or embryo development medium contained the inorganic ingredients of the Emershad and Ramming medium [197] supplemented with 6% sucrose, 4 mM asparagine, and 0.3% activated charcoal. For embryo germination and shoot development, a solid WPM medium [99] supplemented with 2% sucrose, 4 mM asparagine, 0.3% activated charcoal, 5.7 μM IAA, 4.4 μM 6-BAP, and 1.4 μM GA_3_ was used. In the second stage of the ER protocol, the larger healthy seeds that developed from the immature ovules of 8 weeks from were directly plated on a solid WPM medium [99] to induce embryo germination. The rooting medium consisted of a mixture of half-strength MS salts supplemented with 0.3% agar and 1.7 µM IAA. The immature seeds were halved, and the embryo–ovules were embedded with the cut surface in contact with solid soil. Embryo germination was conducted for 8 weeks at 20 ± 2 °C with a 16 h photoperiod. The third stage of the ER protocol involved transferring the young shoots to glass flasks containing a solid rooting medium. For acclimatization, shoots with well-developed roots were transplanted into pots containing a synthetic soil mixture [196]. The results obtained from this protocol allowed the following conclusions: the best conditions were achieved by immature ovules at 40 DAP and inducing embryo germination for 8 weeks; hybrids of the Thompson, Superior, and Regal cultivars showed the highest efficiency of embryo germination; furthermore, the highest percentage of viable plants was obtained from ovules taken at 50 DAP from the progenies of the cross Luisa × Thompson.

**Table 1 plants-12-03106-t001:** Some examples where the embryo rescue (ER) supports introgression of useful traits from wild species or other crops into dicot crops or hybrids.

Crops	Traits	Introgressed from Wild/Crops	References
*Brassica rapa*	^1^ CMS and *Rfp*	*Brassica napus*	[133]
*Brassica napus*	Tolerance to herbicides	*Sinapis arvensis*	[135]
*Brassica juncea*	Tolerance to herbicides	*Brassica kaber* (*S. arvensis*)	[134]
*Brassica juncea*	Tolerance to aphids	*Brassica tournefortii*	[154]
*Brassica juncea*	Tolerance to drought	*Brassica tournefortii*	[154]
*Brassica rapa*	Tolerance to herbicides	*Brassica kaber* (*S. arvensis*)	[134]
*Brassica rapa* subsp. *pekinensis*	Yellow/orange color of petals	*Brassica juncea × B. rapa*	[136]
*Brassica napus*	Clubroot-resistant gene, *CRd*	*Brassica rapa* subsp. *pekinensis*	[139,142]
*Brassica napus*	Resistance to blackleg disease	*Brassica nigra*	[141]
*Brassica oleracea* var. *botrytis*	Resistance to black rot disease	*Brassica carinata*	[146]
*Brassica oleracea*	Resistance to atrazine	*Brassica napus*	[147]
*Brassica oleracea* var. *capitata*	Resistance to *Plasmodiophora brassicae*	*Brassica napus*	[150]
*Brassica oleracea* var. *botrytis*	Resistance to powdery mildew	*Brassica carinata*	[149]
*Brassica oleracea*	Resistance to black rot disease	*Brassica* spp.	[151]
*Brassica oleracea*	Male sterility	*Brassica napus*	[152]
*Brassica oleracea*	CMS *Tour*	*Brassica tournefortii*	[153]
*Phaseolus vulgaris*	Resistance to Ascochyta blight	*Phaseolus coccineus*	[171]
*Phaseolus vulgaris*	Resistance to bean golden mosaic virus	*Phaseolus polyanthus*	[171]
*Phaseolus vulgaris*	Resistance to bean fly	*Phaseolus coccineus*	[171]
*Solanum lycopersicum*	Resistance to tobacco mosaic virus	*Solanum peruvianum*	[173]
*Solanum lycopersicum*	Resistance to tomato spotted wilt virus	*Solanum peruvianum*	[173]
*Solanum lycopersicum*	Resistance to root-knot nematode	*Solanum peruvianum*	[173]
*Carica papaya*	Resistance to papaya ringspot virus	*Carica cauliflora*	[174]
*Vitis vinifera*	Resistance to *Plasmopora viticola*	Chinese wild *Vitis* spp.	[193,195]
*Vitis vinifera*	Seedless	Chinese wild *Vitis* spp.	[194,195]
*Vitis vinifera*	Seedless	*Vitis vinifera*	[196]
*Glycine max*	Resistance to cyst nematode	*Glycine tomentella*	[185]
*Gypsophila paniculata*	Double flowers	*Gypsophila manginii*	[121]
*Gentiana scabra*	Flower color	*Gentiana straminea*	[118]
*Gentiana triflora*	Flower morphology	*Gentiana paradoxa*	[118]
*Gentiana paradoxa* (female)	Petaloid stamen	*Gentiana septemfida* (male)	[118]
*Campanula carpatica*	Flower shape	*Campanula carpatica*	[117]
*Campanula medium*	Tolerance to cold	*Campanula formanekiana*	[117]
*Chrysanthemum × grandiflorum*	Resistance to aphids	*Chrysanthemum przewalskii*	[15,112]
*Chrysanthemum × grandiflorum*	Tolerance to heat	*Chrysanthemum makinoi*	[15]
*Chrysanthemum × grandiflorum*	Tolerance to drought	*Chrysanthemum indicum*	[114]
*Chrysanthemum* × *morifolium*	Tolerance to salt	*Artemisia japonica*	[115]
*Dendranthema crissum*	Heterosis	*Crossostephium chinense*	[116]
*Camellia japonica*	Yellow color of petals	*Camellia chrysantha*	[120]

^1^ CMS—cytoplasm male sterility.

#### 4.1.2. Monocot Species

Bread wheat (*Triticum aestivum*, 2*n* = 6*x* = 42, genome AABBDD) suffers from low genetic variation due to its recent origin from one or a limited number of hybridization events, subsequent domestication, and selection, about 10,000 years ago [198]. Emmer wheat (*Triticum turgidum* subsp. *dicoccum* and *dicoccoides*, 2*n* =  4*x*  =  28, genome AABB), durum wheat (*T. turgidum* subsp. *durum*, 2*n* = 4*x* = 28, genome AABB), *T. timopheevii* (2*n* = 4*x* = 28, genome AAGG), and *Aegilops* species containing the D genome offer excellent sources of genetic variability for bread wheat improvement. In particular, beneficial characteristics, such as tolerance to cold and salt, leaf and stem rust resistance, and resistance to cereal cyst and root-knot nematodes, exist within the allopolyploid *Aegilops* species [199]. However, the potential of these allopolyploids as a source of genetic variability still needs to be explored (reviewed by Mirzaghaderi and Mason [200]). In addition, crosses involving *Aegilops* species often encounter post-zygotic barriers, necessitating ER to develop synthetic wheat lines [201]. Approximately 200 different cross combinations were performed between diverse wheat genotypes and *Aegilops* species, including emmer wheat  ×  *Ae. tauschii* (2*n* =  DD or DDDD), durum wheat  (2*n* = 4*x* = 28, genome AABB) ×  *Ae. tauschii*, *T. timopheevii*  ×  *Ae. tauschii*, *Ae. crassa* ×  durum wheat, *Ae. cylindrica* ×  durum wheat, and *Ae. ventricosa*  ×  durum wheat crosses [199]. However, it is essential to note that in some crosses, such as tetraploid wheat × *Ae. tauschii*, the production of hybrid seeds was low. Therefore, ER was crucial for obtaining viable hybrids. Seeds unable to complete maturation were isolated, sterilized, and soaked overnight in distilled water at 4 °C. The seed coat was removed, and the embryos were cultured on solidified MS medium [72] supplemented with vitamins. The cultures were maintained at 22–24 °C under a 16 h photoperiod. The seedlings were subsequently transferred to soil and brought to maturity. However, although the average outcrossing ability between *T. turgidum × Ae. tauschii* was very low (0.062), more than fifty hexaploid and octaploid F_2_ lineages were successfully obtained through the in vitro rescue of F_1_ embryos and the spontaneous production of F_2_ seeds from F_1_ plants. Notably, crosses such as *T. durum* ×  *Ae. tauschii*, *Ae. crassa* × *T. durum*, *Ae. cylindrica* × *T. durum*, and *Ae. ventricosa* × *T. durum* displayed highly significant morphological and nutritional differences in characteristics, such as the number of tillers and nodes, peduncle length, flag leaf dimension, spikelets per spike, spikelet length, awn length, plant height, and levels of Fe and Zn adsorbed. These genetic resources could be utilized to enhance genetic variation in wheat through crossbreeding and recurrent backcrossing [199].

The productivity of bread wheat is often hindered by both biotic and abiotic stresses. To counter them, continuous efforts are underway to introduce new resistance/tolerance traits into wheat cultivars, taking advantage of the genetic diversity in wheat and related species. The species rye (*Secale cereale*, 2*n* = 2*x* = 14, genome RR) has been extensively used to increase stress resistance in wheat. The first stable amphidiploid *Triticale* (*Triticosecale* Wittmack) was obtained by W. Rimpau in 1888 [202], followed by subsequent attempts to produce wheat–rye hybrids. Homoeologous pairing between rye and wheat allows the introduction of desirable agronomic traits into wheat from rye, including resistance to various biotic and abiotic stresses. Since the 1950s, addition, substitution, and translocation lines have been created from these crosses and have played a significant role in wheat genetic improvements (extensively reviewed by Crespo-Herrera et al. [203]).

The incorporation of various rye genes into wheat confers resistance against fungal pathogens, such as leaf rust (*Puccinia triticina* Erikss.), yellow rust (*Puccinia striiformis* var. *striiformis* Westend), stem rust (*Puccinia graminis* Pers. f. sp. *tritici* Erikss. and E. Henn.), and powdery mildew (*Blumeria graminis* (DC.) f. sp. *tritici* Em. Marchal). Additionally, rye genes confer resistance to some of the most important grain pests, including aphid species such as *Schizaphis graminum* (Rondani), *Diuraphis noxia* (Mordvilko), *Rhopalosiphum padi* L., and *Sitobion avenae* (F.); the cecidomyid *Mayetiola destructor* (Say); the nematodes *Heterodera avenae* (Wollenweber) and *Heterodera filipjevi* (Madzhidov) Stelter; and the mite *Aceria tosichell* Keifer. Furthermore, the tolerance of rye to abiotic stresses, like aluminum toxicity and acidic soils, can be utilized in wheat improvement efforts [203].

An et al. [204] developed a crossbreeding program between *Triticum aestivum* and *Secale cereale* to create addition lines of wheat with the rye chromosome 6, which possesses the *Pm20* gene on its long arm (Table 2).

The *Pm20* gene confers powdery mildew resistance. Twelve plants were obtained from immature embryos cultured on MS medium [72]. Six 2*n* = 6*x* = 56 amphidiploid plants were synthesized following chromosome complement doubling with colchicine. After powdery mildew resistance screening, the six plants were backcrossed to wheat. BC_1_F_1_-resistant plants were subsequently used as female plants and crossed continuously with wheat. Powdery mildew resistance tests and karyological analyses were performed on BC_2_F_1_ plants. Resistant plants of putative monosomic addition lines with chromosome number 2*n* = 43 were self-pollinated. The plant with chromosome number 2*n* = 44 was selected and continuously self-pollinated for five generations. During this breeding process, each plant was selected for resistance to powdery mildew, morphological similarity with wheat, and the high number of seeds produced. Finally, a fertile and genetically stable line with homogeneous resistance, designated WR49-1, was selected. Molecular analyses, including GISH (genomic in situ hybridization), mc-FISH (multicolor fluorescence in situ hybridization), mc-GISH (multicolor-GISH), and EST (expressed sequence tag) showed that WR49-1 represented a new wheat–rye 6R disomic addition line. WR49-1 adult plants displayed high levels of resistance to wheat powdery mildew (*Blumeria graminis* f. sp. *tritici*, *Bgt*). Notably, young WR49-1 seedlings showed high resistance levels against 19 out of 23 *Bgt* isolates. Therefore, WR49-1 seedlings could have novel powdery mildew resistance genes distinct from *Pm20* on the 6RL chromosome arm of rye. Because of its cytological stability and agronomic traits of interest, WR49-1 holds promise as a bridging addition line in wheat breeding. Recent research has suggested that the region on the long arm of rye chromosome 6, called YT2, likely harbors genes conferring powdery mildew resistance in a different chromosomal region than *Pm20* [205]. These findings are significant for seeking new sources of resistance, especially against fungal pathogens, because their rapid genetic variation often allows them to overcome resistance in recently improved varieties. A recommended strategy is gene pyramiding to enhance resistance durability, which involves incorporating resistance against multiple fungal strains [206].

An et al. [207] performed a *Triticum aestivum* × *Secale cereale* cross to evaluate resistances to some pathogens. Hybrid embryos of 16–18 DAP were isolated and cultured on MS medium [72]. Seventeen mature plants were obtained. Chromosome complement doubling with 0.05% colchicine led to the generation of nine amphidiploid plants (2*n* = 6*x*= 56). These amphidiploids were tested for resistance against *Blumeria graminis* f. sp. *tritici* (*Bgt*) isolates and a race mixture of *Puccinia striiformis* f. sp. *tritici* (*Pst*). The resistant progenies were backcrossed with bread wheat, resulting in the development of a line named WR35, which exhibited resistance to both powdery mildew and stripe rust. The authors employed sequential GISH, mcFISH, and ND-FISH (non-denaturing FISH) techniques with multiple probes, mc-GISH, analysis of rye chromosome arm-specific markers, and SLAF-seq (sequencing of amplified fragments of specific locus) to demonstrate that WR35 was a disomic 4R wheat–rye line (2*n* = 44). WR35 showed cytological stability and high productivity. At the adult stage, WR35 displayed high resistance to powdery mildew (*Bgt*), stripe rust (*Pst*), and *Rhizoctonia cerealis*. Moreover, at the seedling stage, WR35 exhibited resistance to twenty-two *Bgt* isolates and four *Pst* races, indicating its resistance to multiple diseases. An et al. [207] suggested that WR35 could represent a promising parent for wheat chromosome engineering (Table 2).

In addition to traditional breeding approaches, character transfer from rye to wheat can also be undertaken through in vitro culture of immature embryos from the wheat–rye hybrid (genome AABBDDRR) [208]. In fact, it is well known that in vitro culture, particularly when a prolonged callogenesis phase precedes the development of adventitious shoots and/or embryos, can induce a wide range of genetic and epigenetic modifications. In regenerated plants, these include chromosomal instability, leading to changes in chromosome number, chromosome breaks, deletions, translocations often resulting from the activation of transposable elements, and gene mutations [209]. Lapitan et al. [208] cultured in vitro immature (11–13 DAP) wheat–rye hybrid embryos. Calli that originated from two embryos were cultured for 222 days in media supplemented with 0.5 mg L^−1^ of 2,4-D. The regenerated plantlets were transplanted and cloned. To induce amphidiploids, plants were backcrossed with wheat or treated with colchicine. Through this process, wheat–wheat and wheat–rye chromosome translocations, deletions, and amplifications of heterochromatin bands of rye chromosomes were identified. These results suggested a high degree of structural changes induced by tissue culture in the chromosomes of wheat and rye. Therefore, tissue culture may be a useful tool in alien gene introgression in both wheat and *Triticale* improvement.

Wild barley (*Hordeum bulbosum*) exhibits resistance to powdery mildew (*Erysiphe graminis* D.C. f. sp. *hordei* Marchal), whereas winter barley (*H. vulgare* and *H. sativum*) is susceptible to the infection. Interspecific hybrid plants resistant to powdery mildew were obtained through the ER of hybrid embryos [210]. Subsequently, powdery mildew resistance was introgressed into *H. vulgare* by *H. bulbosum* by crosses between four diploid barley cultivars and a landrace of tetraploid *H. bulbosum* [211] (Table 2). The embryos were cultured in B5 medium [71] without PGRs. The dissection of embryos was performed at 8–10 DAP. The hybrids followed by backcrosses to *H. vulgare* were performed, and molecular mapping revealed the translocation of chromosomal regions from *H. bulbosum* containing the powdery mildew resistance. In addition, the segregation observed in BC_2_ plants also suggested the presence of a single dominant gene responsible for the transferred resistance [211].

In the genus Asparagus, there are several wild relatives of *A. officinalis*, including *A. densiflorus*, *A. macrorrhizus*, *A. prostratus*, *A. maritimus*, *A. breslerianus*, *A. oligoclonos*, *A. pseudoscaber*, *A. brachyphyllus*, *A. tenuifolius*, *A. persicus*, *A. verticillatus*, *A. kasakstanicus*, and *A. kiusianus*. These wild species harbor significant genetic variability and could serve as valuable genetic tools for the development of new varieties through interspecific hybridization (reviewed by Encina and Regalado [212]). Unfortunately, reproductive barriers, both pre- and post-zygotic, have hindered successful hybridization between *A. officinalis* and these wild species with potential breeding value. To circumvent this barrier, in vitro ER using ovule and ovary cultures has been tested in the *A. officinalis* × *A. densiflorus* cross [213]. *A. densiflorus* has resistance to several diseases caused by pathogens, such as *Fusarium oxysporum* f. sp*. asparagi*, *Fusarium moniliforme*, *Fusarium proliferatum*, and *Fusarium solani* [214,215]. However, despite the cultivation of 2032 ovules and 826 ovaries in vitro, no seedlings were obtained from the *A. officinalis* × *A. densiflorus* cross [213]. Histological studies revealed only protoderm differentiation, endosperm collapsed, and premature embryonic abortion.

In the genus *Alstroemeria*, interspecific hybridization followed by ER has been widely used as a breeding technique to obtain new varieties for the flower market. Winter-hardy hybrids were developed by using the Chilean species *Alstroemeria aurea*, while fragrant hybrids were generated using the Brazilian species *Alstroemeria caryophyllaea* (Table 2). The successful production of interspecific hybrids was promoted by the in vitro ovule culture [216]. Ovules were harvested at 10–23 DAP and cultured on 1/4 MS medium [72] under continuous dark conditions until germination. Seedlings were subcultured on basal MS medium [72] supplemented with 2 mg L^−1^ Benzyladenine (BA) until they reached sufficient size for rooting. After profuse root development, the plants were transferred to the greenhouse for acclimatization. Hybrids were evaluated both in the greenhouse and in the field. Noteworthily, these efforts resulted in new cultivars characterized by winter hardiness, blooming growth habits, and long flower stems [216]. Recently, a study reported the successful rescue of 3669 embryos from extensive interspecific hybridization within the genus *Alstroemeria*. Eighteen plants were evaluated morphologically, showing interesting characteristics for the ornamental plant market [217] (Table 2).

The African oil palm (*Elaeis guineensis* Jacq.) is native to Africa and is currently distributed in Africa, Southeast Asia, and Central and South America. In Central America, oil palm plantations are affected by a wide variety of pests and diseases, which significantly reduce productivity. Fatal yellowing (FY) is the most damaging phytoplasma disease among them. Genetic resistance to this disease can be found in the germplasm of the American oil palm (*E. oleifera*), a native species from the Amazon rainforest [94]. However, the seeds obtained from the cross *E. guineensis* and *E. oleifera* had lower germination rates. To address this issue, an in vitro system for germinating hybrid embryos was developed. MS medium [72] supplemented with 110 mM glucose was the most effective. The emergence for shoots and rootlets was 97% and 73%, respectively. The survival rate of the seedlings 45 days after transplanting into the soil was 55%, and by the 4th month, 66 plants had produced sufficiently developed shoots and roots. Therefore, the method is satisfactory for transferring FY resistance of *E. oleifera* to *E. guineensis* [94].

In the Colchicaceae, intergeneric hybrids were produced through reciprocal crosses between *Sandersonia aurantiaca* and *Littonia modesta* [218] to generate new genotypes with improved features. The embryos were rescued by culturing ovules ranging from 14 to 30 DAP. The ovules were cultured on solidified KM medium [219] with salts, microelements, and vitamins B5 [71]. Furthermore, the medium was supplemented with 20 mL L^−1^ coconut water, 10 g L^−1^ sucrose, and 250 g L^−1^ casein hydrolysate. Cultures were maintained at 24 ± 2 °C under a 16 h photoperiod. Germinated embryos were cultured on solidified medium with half-strength MS salts [72], vitamins B5, and 10 g L^−1^ sucrose. The seedlings had grown to bulb production. Chromosome counts of *S. aurantiaca, L. modesta*, and the hybrids revealed chromosome numbers of 2*n* = 24, 22, and 23, respectively. Consequently, the hybrids were sterile, but their survival could be sustained by agamic reproduction [218]. Amano et al. [220] conducted an intergeneric hybridization program using six genotypes of *Gloriosa* spp., one genotype of *L. modesta*, and two genotypes of *S. aurantiaca* to increase genetic variability and obtain new cultivars. Through the ER of ovules, hybrids were obtained from several crosses: *L. modesta* × *S. aurantiaca*, *L. modesta* × *S. aurantiaca* “Phoenix”, *L. modesta* × *G. superba* “Lutea”, *L. modesta* × *G*. *superba* “Marron Gold”, *S. aurantiaca* × *G. superba* “Lutea”, and *S. aurantiaca* × *G.* “Marron Gold”. More than 3800 ovules were obtained from 32 intergeneric cross combinations. Ovules with placental tissues were cultured on solidified half-strength MS medium [72] supplemented with 0.01 mg L^−1^ NAA and 0.01 mg L^−1^ BA at 25 °C in the dark [221]. Rhizome-like structures were differentiated from ovules in 30 of 32 cross combinations. For shoot regeneration, ovules with rhizome-like structures were transferred to half-strength MS medium containing 0.25 mg L^−1^ NAA and 2.5 mg L^−1^ BA at 25 °C with a 16 h photoperiod. For rooting, the shoots were transferred to half-strength MS medium without PGRs. Rooted seedlings were able to produce small tubers that could be grown in soil. Amplified fragment length polymorphism (AFLP) analyses confirmed the hybrid nature of the obtained plants. Many of these crosses showed interesting morphological traits for the flower market, and some of these characteristics are listed in Table 2.

In *Lilium* spp., which comprises approximately 100 known species, interspecific hybridization has been widely used since the 1800s in breeding programs to produce attractive hybrids for the flower market (reviewed extensively by Marasek-Ciolakowska et al. [55,83]). Many characteristics have been introduced into commercial hybrids, including a short juvenile phase, early flowering, pink flower color, upright flowers, and yellow flower color. In addition, considering the enormous variability in many species in the genus *Lilium*, many other characteristics, such as those related to floral symmetry and the number and shape of corolla organs, can be introduced into commercialized species. One promising cross is between Oriental lilies and several species such as *L. henryi*, Asiatics, and trumpet lilies. However, both pre- and post-zygotic barriers can hinder hybridization in many cases, particularly between phylogenetically more distant species. Pre-zygotic barriers are efficiently overcome using the cut-style pollination technique, while ER techniques can be valuable for addressing post-zygotic barriers [55,83]. For the success of the ER in interspecific hybridization of the genus *Lilium*, several factors must be considered. Mainly, it is necessary to determine the direction of the crossing because unilateral incompatibility is often observed. The selection of genotype, both of the female and the male plant, is crucial for a successful cross. As in many other genera, a preliminary study is needed to identify the optimal DAP for the isolation of immature hybrid embryos. Several ER techniques are available, including the culture of embryos, ovules, and ovaries. With regard to the culture conditions, MS medium [72] was found the most suitable for the culture of immature embryos. It is necessary to obtain a high survival and germination rate of the hybrid embryos using media with a relatively sub-acidic pH (pH 5.0). Likewise, a medium supplemented with 20–40 g L^−1^ of sucrose and with a concentration of NAA considerably high (10^−2^–10^−4^ mg L^−1^) is often required. Generally, very small embryos need high concentrations of sugar. In general, embryo culture of *Lilium* spp. requires an optimal sucrose concentration of 6% or 3% sucrose plus 2% mannitol.

Kim et al. [65] successfully generated interspecific hybrids between two lily species, *Lilium hansonii* Leichtlin and *L. brownie* var. *colchesteri*, using conventional pollination and cutting-style methods. However, viable seeds were only obtained when *L. hansonii* was used as the female parent. These two species significantly differ in terms of biotic stress resistances and floral characteristics, making them promising candidates for obtaining interesting materials for the flower market. After the crosses, fruits were harvested at 60 DAP, and the seeds were collected and sterilized. The seeds were cultured on solidified MS medium supplemented with 30 g L^−1^ sucrose and 0.1 g L^−1^ activated carbon. Germination of the seeds occurred at 23 °C under a 12 h photoperiod. For immature seeds, the hybrid embryos were grown on the same culture medium for three weeks until the embryo reached a size of about 2~3 mm. Subsequently, the embryos were transferred to a fresh MS medium until they fully matured. Ninety-two hybrids were confirmed by SSR marker analysis. The hybrids exhibited distinct morphological features such as pollen and stigma color, plant height, flower diameter, the number of shoots, tepal size, and flower direction [65] (Table 2).

ER has been widely used to overcome post-fertilization barriers in interspecific crosses in the genus *Tulipa* (reviewed by Marasek-Ciolakowska et al. [55,83]). Interspecific crosses, *Tulipa gesneriana* × *T. kaufmanniana* Regel*, T. gesneriana* × (*T. kaufmanniana* Regel × *T. greigii* Regel), and *T. gesneriana* × (*T. fosteriana* Hoog × *T. kaufmanniana* Regel), were performed by Custers et al. [64]. The crosses aimed to obtain new hybrids with novel floral characteristics in both the color and shape of the floral organs. From unripe seed pods, ovules were collected, and in some cases, embryos were isolated. A single modified MS medium [72] was used to culture the ovules and immature embryos. The solidified medium was supplemented with 4% sucrose, 500 mg L^−1^ tryptone, and 4 μM NAA. To compare the culture efficiency of isolated embryos with that of ovules, the cultures were started at different DAPs, also in relation to the interspecific cross. The environmental conditions of cultivation are fundamental for bulbous plants such as the tulip [53,83,84]. The cultures were kept at 15 °C in the dark until 15 weeks after pollination. A treatment at 5 °C for 12 weeks was carried out to induce embryo germination. To obtain bulbils, seedlings were grown for 12–18 weeks at 15 °C. Maturation was carried out for 6 weeks at 22 °C. The bulbils were grown in soil at 5 °C for three months and transferred to 15 °C for germination. For efficient embryo retrieval, the time of collection was crucial. Optimal ER was reached in cultures that started 7 to 9 weeks after pollination. In addition, using ovule cultures significantly increased the efficiency of seedling formation. Interesting floral characteristics were observed in the hybrids, such as early flowering, dark base color of the corolla, long and pointed petals, and petals with a carmine-red dorsal band [64] (Table 2).

**Table 2 plants-12-03106-t002:** Some examples where embryo rescue (ER) supports introgression of useful traits from wild species or other crops into monocot crops or hybrids.

Crops	Traits	Introgressed from Wild/Crops	References
*Alstroemeria* hybrid	Winter hardiness	*Alstroemeria aurea*	[216]
*Alstroemeria* hybrid	Fragrant flowers	*Alstroemeria caryophyllea*	[216]
*Alstroemeria* hybrid	Ever-blooming growth habit	*Alstroemeria caryophyllea*	[216]
*Alstroemeria* hybrid	Long flower stems	*Alstroemeria caryophyllea*	[216]
*Alstroemeria* hybrid	Flower color and morphology	*Alstroemeria pelegrina*	[217]
*Alstroemeria* hybrid	Flower color and morphology	*Alstroemeria caryophyllea*	[217]
*Alstroemeria* hybrid	Fragrant flowers	*Alstroemeria caryophyllea*	[217]
*Hordeum vulgare*	Resistance to powdery mildew	*Hordeum bulbosum*	[210,211]
*Triticum aestivum*	Resistance to powdery mildew	*Secale cereale* (addition line ^1^ ch. 6R)	[204,205]
*Triticum aestivum*	Multiple fungal disease resistance	*Secale cereale* (addition line ^2^ ch. 4R)	[207]
*Elaeis guineensis*	Resistance to fatal yellowing	*Elaeis oleifera*	[94]
*Littonia modesta*	Climbing habit	*Sandersonia aurantiaca*	[220]
*Littonia modesta*	Color of corolla	*Sandersonia aurantiaca*	[220]
*Littonia modesta*	Climbing habit	*Gloriosa* spp.	[220]
*Littonia modesta*	Pendulous flowers	*Gloriosa* spp.	[220]
*Littonia modesta*	Undulate and slightly reflexed tepals	*Gloriosa* spp.	[220]
*Littonia modesta*	Modification of corolla color	*Gloriosa* spp.	[220]
*Sandersonia aurantiaca*	Pendulous flowers	*Gloriosa* spp.	[220]
*Sandersonia aurantiaca*	Undulate tepals fused at the base	*Gloriosa* spp.	[220]
*Tulipa gesneriana*	Dark basic color of corolla	*Tulipa kaufmanniana*	[64]
*Tulipa gesneriana*	Early flowering time	*Tulipa kaufmanniana*	[64]
*Tulipa gesneriana*	Long pointed petals	*Tulipa kaufmanniana*	[64]
*Tulipa gesneriana*	Petals with carmine-red dorsal band	*Tulipa kaufmanniana*	[64]
*Lilium hansonii* (female)	Increase in flower spots	*L. brownii* var. *colchesteri* (male)	[65]
*Lilium hansonii* (female)	Increase in flower size	*L. brownii* var. *colchesteri* (male)	[65]
*Lilium hansonii* (female)	Increase in number of buds	*L. brownii* var. *colchesteri* (male)	[65]
*Lilium hansonii* (female)	Tepal size	*L. brownii* var. *colchesteri* (male)	[65]
*Lilium hansonii* (female)	Reduction in plant height	*L. brownii* var. *colchesteri* (male)	[65]
*Lilium hansonii* (female)	Color of pollen and stigma	*L. brownii* var. *colchesteri* (male)	[65]
*Lilium hansonii* (female)	Flower orientation	*L. brownii* var. *colchesteri* (male)	[65]

^1^ ch. 6R—chromosome 6 of rye (*Secale cereale* L.); ^2^ ch 4R—chromosome 4 of rye.

### 4.2. Embryo Rescue to Recover Haploids, Dihaploids, Doubled Haploids, and Polyploid Embryos

Interspecific and intergeneric crosses, in vitro culture of male (androgenesis) and female (gynogenesis) gametophytes, provide, for many species, powerful tools for manipulating ploidy to facilitate the selection and development of new crops. In addition, interspecific and intergeneric hybridizations are critical for generating disomic alien chromosome addition lines and alien chromosome substitution lines relevant for genetic research and chromosome engineering.

Haploid plants, characterized by a gametic number of chromosomes (*n*), play a significant role in plant improvement programs. They are fundamental in many research disciplines, including biotechnology, cytogenetics, and plant breeding. In fact, homozygous DH genotypes can be obtained with natural or artificial chromosome doubling of haploids.

Although some authors suggest that haploids occur infrequently in normal cultivated species under experimental conditions, there are genetic–physiological mechanisms that account for their spontaneous occurrence. For instance, spontaneous haploids (2*n* = 2*x* = 14) have been reported in *Triticum durum* (2*n* = 4*x* = 28) [222]. Haploids may be spontaneously produced by occasional parthenogenetic development of gametic chromosome-numbered egg cells. Under natural conditions, haploid parthenogenesis may result from a sudden heat shock that temporarily halts or delays pollen tube growth, allowing the auxin factors produced by the pollen tube to stimulate oosphere development without fertilization [223]. Moreover, in species with ovaries containing numerous ovules, such as legumes, Solanaceae, and orchids, the simultaneous penetration of the ovary by multiple pollen tubes can lead to an induction of auxin stimulus. This stimulus can trigger the parthenogenetic development of oospheres that have not yet been reached by the pollen tube. In other cases, alongside the amphimictic development of the oosphere, a synergid can form a haploid embryo, producing a two-embryo seed (haploid-diploid twins) [223].

Androgenesis is the principal method to obtain haploids through in vitro cultures of anthers and isolated microspores. Guha and Maheshwari [224] demonstrated for the first time that when *Datura innoxia* anthers (2*n* = 2*x* = 24) were grown in vitro under special nutritional conditions, the pollen grains were stimulated to undergo mitosis, and the resultant embryos developed haploid plants (*n* = 12). Androgenesis has been successfully applied in many species (reviewed by Portemer et al. [225]; Ren et al. [226]; and Hale et al. [227]). Haploids can be also obtained through gynogenesis from unfertilized egg cells or synergids (reviewed by Portemer et al. [225]). San Noeum [228] achieved the first successful production of haploid plants from non-pollinated ovary or ovule cultures in barley (*Hordeum vulgare*, 2*n* = 2*x* = 14).

Haploids have also been obtained through the pollination of distant species, belonging to different species or genera. Joergensen [229], many years ago, obtained haploids of *Solanum nigrum* by pollinating flowers of this species with pollen from *S. luteum*. He showed that in these crosses, pollination is followed by regular pollen tube growth and fertilization of the secondary nucleus of the female gametophyte, leading to albumen formation. Meanwhile, the other sperm nucleus degenerates into the cytoplasm of the oosphere, which can induce parthenogenetic development. Interspecific and intergeneric crosses have resulted in maternal haploids in several species. However, there are also known cases of the development of androgenetic haploids due to the degeneration of the oosphere nucleus, which is then replaced by a sperm nucleus. Therefore, cells containing the paternal genome in a maternal cytoplasm have been obtained [230].

Several other methods have been adopted to obtain haploid plants, for example, by delaying pollination. In this case, flowers are emasculated, and stigmas are pollinated with a delay; in this way, egg cells can be stimulated to divide before fertilization [231].

The irradiation of female gametophytes can induce androgenetic haploids. Gerassimova [232] first demonstrated this possibility by pollinating, with pollen from a *Crepis tectorum* plant, which is homozygous for some recessive genes, another plant of the same species, which is homozygous for the same genes in the dominant state, that had been subjected to X-irradiation; hence, a haploid induced brought the characteristics determined by the recessive alleles. The irradiation of male gametophytes is another method used to obtain gynogenetic haploids, based on earlier experiments by Goodspeed and Avery [233] on *Nicotiana glutinosa* (2*n* = 2*x* = 24). Subsequently, it has been widely used. Interestingly, high-dose irradiation of pollen renders the sperm nuclei incapable of fertilization due to lethal mutations. However, it regularly develops the pollen tube by providing the auxin stimulus to egg cell parthenogenesis. In Poaceae, Kihara and Tsunewaki [234] developed a method called “alien cytoplasm,” which involves the replacement of the nucleus into a cytoplasm of another species. They applied this method to *T. aestivum* and *Triticale*, using *Ae. caudata* (2*n* = 2*x* = 14, genome CC) cytoplasm (alien cytoplasm). Plants of *Ae. caudata* were pollinated with pollen from *T. aestivum* var. *erytrospermum*, and the hybrids were backcrossed for thirteen generations, which resulted in the establishment of substitution lines of *T. aestivum* in the cytoplasm of *Ae. caudata*. Furthermore, these substitution lines were pollinated with *Triticale* (2*n* = 6*x* = 42) pollen, and the F_1_ was backcrossed with *Triticale* twice, resulting in *Triticale* with *Ae. caudata* cytoplasm. The authors demonstrated that the substitution lines of *T. aestivum* had a frequency of 1.7% haploids, while *Triticale* with alien cytoplasm had as many as 53% haploids. Thus, it was demonstrated that an alien cytoplasm (*Ae. caudata*) enhances the tendency of a species to undergo haploid parthenogenesis, and different genotypes respond differently to the same alien cytoplasm [234].

From 1947 onward, C.L. Huskins and collaborators showed that low-dose colchicine and other mitotic poisons are sometimes capable of bringing about dividing cells, the separation in the prophase of chromosomes into two groups—sometimes haploid and homologous—each organizing its spindle. This reductional grouping process can generate four haploid cells from a diploid somatic cell [235]. The occurrence of somatic chromosomal reduction as a mechanism for haploid formation was demonstrated by Simantel et al. [236] in *Sorghum* species treated with colchicine, leading to the production of haploids from diploid plants and dihaploids from tetraploid plants. Somatic chromosome reduction may occasionally occur during development, as observed in *Haplopappus gracilis* (2*n* = 2*x* = 4) [237].

An outstanding opportunity for haploid generation is through the induction of parthenogenesis triggered by the *BABY BOOM* gene (reviewed by Kalinowska et al. [238]). This method has been demonstrated in pearl millet [239], maize, rice [240], and tobacco [241]. The support of embryo cultures at various stages of this methodology is very interesting.

Examples of ploidy level variations obtained in dicot and monocot species using different methodologies are presented in Table 3 and Table 4, respectively.

#### 4.2.1. Dicot Species

Haploid embryos of *Cichorium intybus* L. (2*n* = 2*x* = 18) were obtained a few days after pollination with *Cicerbita alpina* Walbr. (2*n* = 2*x* = 18) [242] (Table 3).

Extensive breeding programs based on this type of hybridization require very effective methodologies for ER and ploidy evaluation (Figure 1). Haploid embryos of lettuce (*Lactuca sativa*, 2*n* = 2*x* = 18) were successfully developed through in vivo distant pollination with fresh pollen grains of *Helianthus annuus* L. (2*n* = 2*x* = 34) or the hexaploid *H. tuberosus* L. (2*n* = 6*x* = 102) [243]. Although the haploid proembryos obtained after pollination did not progress further, 23 haploid plantlets were successfully grown from embryo sacs isolated using ER [243]. Ovaries at 5–10 DAP with ovules and globular embryos were isolated. Ovules were cultured on 28 combinations of MS media [72,243] in the dark at 22 °C for seven days and then transferred to the light at 16 °C. Callus proliferated from globular embryos on only two media: (i) MS supplemented with 2 mg L^−1^ BAP and 1 mg L^−1^ IAA; (ii) MS supplemented with 1 mg L^−1^ NAA and 1 mg L^−1^ 2,4-D. Calli were transferred in regeneration MS medium supplemented with 2 mg L^−1^ kinetin for four weeks. Well-developed rooted plantlets were transferred to pots. Genome size by flow cytometry and chromosome counting in root tips demonstrated the haploid nature of lettuce.

Ploidy changes significantly affect various traits of ornamental plants, such as flower symmetry and architecture. Haploids, dihaploids, DHs, and polyploids, have been developed in many ornamental plants (reviewed by Mehbub et al. [29]). For example, crosses between tetraploid (2*n* = 4*x* =28) rose (*Rosa × hybrida*) cultivars and other pentaploid or hexaploid rose species in the section Caninae have been obtained, as well as triploid and tetraploid progenies [244] (Table 3).

The methods of interspecific hybrid production in *Primula* involve four steps: (i) emasculation; (ii) pollination; (iii) ER culture; and (iv) confirmation of hybridity and ploidy level of regenerated plants [245]. In particular, interspecific hybrids were obtained in the reciprocal crosses between *Primula filchnerae* (2*n* = 2*x* = 24) and *P. sinensis* “Fanfare” (2*n* = 2*x* = 24) through in vitro ovules recovery [246]. Ovules were cultured on solidified half-strength MS medium [72] supplemented with 50 g L^−1^ sucrose, 0.1 mg L^−1^ NAA, 0.1 mg L^−1^ BA, and 50 mg L^−1^ GA_3_. Embryo rooting was achieved, but SAM development was inhibited. Root systems maintained on the same medium proliferated by producing callus. When calli were transferred in half-strength MS media containing 30 g L^−1^ sucrose without PGRs or with 1 mg L^−1^ zeatin and 0.1 mg L^−1^ NAA, shoot regeneration was induced. Diploid hybrids were obtained when the diploid *P. filchnerae* was used as the maternal parent, while hexaploid hybrids were generated when *P. sinensis* was used as the maternal parent. According to Amano et al. [246], the hexaploid hybrid could be created by chromosome doubling of a triploid obtained when *P. sinensis* was fertilized with unreduced gametes of *P. filchnerae* (Table 3).

Gynogenesis was investigated in three vine cactus species: allotetraploid (2*n* = 4*x* = 44) *Selenicereus megalanthus*, diploid (2*n* = 2*x* = 22) *Hylocereus polyrhizus*, and *Hylocereus undatus* [247]. Unpollinated ovules from developing flower buds containing microspores at the middle uninucleate developmental stage were cultured in vitro. Immature ovules were placed on MS medium [72] supplemented with 0.5 mg L^−1^ 2,4-D, 0.5 mg L^−1^ Thidiazuron (TDZ), and three concentrations of sucrose 0.09, 0.18, and 0.26 M. Cultures were incubated in the dark at 24 ± 2 °C. The enlarged ovules were cultured in MS medium supplemented with 0.2 mg L^−1^ 2,4-D, 0.5 mg L^−1^ TDZ, and 0.09 M sucrose at 24 ± 2 °C with a 16 h photoperiod. Putative gynogenic embryos developed green shoots. Elongated seedlings were hardened in a greenhouse. Ploidy levels were determined for 29 *S. megalanthus* gynogenic plants: 15 DHs (plants with the gametophytic chromosome number) and 14 with higher ploidy levels [247]. ER at a very early developmental stage following interspecific–interploidy crosses between self-incompatible *Hylocereus* spp. was developed [248]. Hand pollinations were performed using the tetraploid (2*n* = 4*x* = 44) *H. megalanthus* as the female parent and either diploid (2*n* = 2*x* = 22) *H. monacanthus* or *H. undatus* as the male parent. Analysis of ploidy levels in 77 putative hybrids revealed diploid, triploid, tetraploid, and higher than tetraploid levels. AFLP analysis was conducted on 22 randomly chosen hybrid progenies, confirming their status as true hybrids [248].

The production of haploids and DHs through unfertilized ovule culture was examined in 19 wild species of gentians (*Gentiana* ssp.) [249]. Unfertilized flower buds were stored at 4 °C in the dark. After sterilization of the flower buds, the ovules were taken from the pistils and cultured on half-strength solidified NLN medium [250] supplemented with 10% sucrose. The cultures were maintained at 25 °C in the dark. Embryonic structures developed from the ovules were transferred to a half-strength MS medium solidified [72] supplemented with 3% sucrose and 1 mg L^−1^ GA_3_. The cultures were incubated at 20 °C with a 16 h photoperiod. After acclimatization, the regenerated plants were transferred to soil and grown in a greenhouse. Among the 19 species, 15 produced regenerated plants from unfertilized egg cultures, and 11 produced embryo-like structures. The ploidy levels of 117 randomly chosen plants out of 333 regenerated were determined, indicating that the majority were diploid or haploid [249]. Unfortunately, SSR analysis showed that diploid plants were not DHs, necessitating chromosome doubling of haploids by treatment with colchicine or other mitotic spindle poisons.

Interspecific hybrids in begonias were successfully obtained through reciprocal crosses between diploid and tetraploid cultivars of *Begonia semperflorens* (genome SS and SSSS) and *B.* “Orange Rubra” (section *Gaerdita* × section *Pritzelia*, genome RR), with the support of in vitro culture of mature or immature embryos [251]. Haploid, DH, triploid, tetraploid, and hexaploid plants with various genomic combinations were generated. The results suggested that unreduced gametes and spontaneous chromosome doubling contribute to the establishment of hybrids. These hybrids exhibited intermediate traits between parents based on their genomic constitutions, and some of them displayed key characteristics from both parents, which are significant for the flower market [251].

DHs of diploid (2*n* = 2*x* = 30) carnation (*Dianthus caryophyllus* L.) were obtained through pseudo-fertilized ovule culture. The emasculated flower buds of carnation were pollinated with pollen inactivated by X-ray irradiation [252]. The ovaries were explanted and cultured in vitro. Cytological analysis of the roots in regenerated plants revealed chimeric situations, with cells having either 15 or 30 chromosomes. DHs were obtained through self-fertilization of the regenerated plants, likely resulting from spontaneous chromosome duplication events.

Ishizaka [253,254] extensively reviewed interspecific hybridization in the genus *Cyclamen* used to obtain new cultivars with valuable morphological traits (e.g., new flower color and fragrant blooming). The recovery of fertility through chromosome doubling of the sterile hybrid to obtain fertile amphidiploids was also analyzed [253]. In addition, the aseptic in vitro culture of placenta-attached ovules containing hybrid embryos allowed the production of allotriploid and allotetraploid plantlets, which were subsequently grown to mature plants in a greenhouse [254]. For instance, an interspecific hybridization between the diploid species *C. persicum* (2*n* = 2*x* = 48) and *C. hederifolium* (2*n* = 2*x* = 34) resulted in a sterile interspecific hybrid with a delay in corolla senescence. However, there was also an interest in obtaining amphidiploids through colchicine treatment. Fertile plants with 82 somatic chromosomes were considered amphidiploids (allotetraploids), resulting from the chromosomal doubling of allodiploids [255] (Table 3). By recovering fertility, this material could serve as a starting point for the establishment of new varieties [255]. Interesting features derived from *C. hederifolium* were transferred into these amphidiploids, such as cold hardiness, flower fragrance, and more attractive leaves [254]. Ishizaka [253] described obtaining other amphidiploids in the *Cyclamen* genus and the introgression of key characteristics into *C. persicum* from wild species [256,257]. Some crosses are reported in Table 3.

The oil produced by seeds of *Jatropha curcas* L. (2*n* = 2*x* = 22) is highly valued in the biodiesel industry. However, the cultivation of *J. curcas* is more complex compared to the closely related species *Ricinus communis* L. (2*n* = 2*x* = 20). Intergeneric hybrids (i.e., *J*. *curcas* × *R. communis*) were obtained to evaluate their agronomic value [258]. Immature hybrid embryos were rescued in vitro, and the hybridity nature of plantlets was confirmed using RAPD markers. Regarding morphological characteristics, such as the number of branches, stem diameter, and leaf size, the hybrids resembled those of *J. curcas*, while the plant height resembled that of *R. communis*. The calli of intergeneric hybrids were obtained in vitro from the young stems using colchicine to induce polyploidy cells. Appropriately, the authors suggested that a high percentage of tetraploid cells (>50%) was crucial in overcoming the sterility problem following hybridization [258]. However, the production of amphidiploid plants has not been reported.

Since 1982, the ER technique has been widely applied to embryo germination of stenospermic grapes in crossbreeding programs. In particular, crosses between grapes of different ploidy levels effectively obtain new seedless cultivars. New hybrid triploid germplasms were obtained using the well-established ER method from five crosses between tetraploid and diploid grape varieties [259]. In grapes, the hybridization program included the following: (i) cross-breeding between seedless *Vitis vinifera* cultivars and wild Chinese *Vitis* spp.; (ii) crossing with seedless cultivars; and (iii) hybridization between grapes of different ploidy levels [260,261]. New seedless lines were obtained using ovule culture [260]. Furthermore, hybrid progenies were obtained by crossing hybrid seeds of the tetraploid (2*n* = 4*x* = 78) *V. vinifera* × *V. labrusca* with the diploid (2*n* = 2*x* = 38) *Vitis vinifera*. Flow cytometry analysis suggested the presence of haploid and triploid plants, although cytological analysis did not confirm these findings on meristematic apices or specific chromosomal markers [260] (Table 3).

In Brassicaceae, Matsuzawa et al. [262] proposed interspecific and intergeneric hybridizations to develop synthetic amphidiploid lines, alien gene introgression lines, alloplasmic lines, monosomic alien chromosome addition lines, and monosomic alien chromosome substitution lines. The first step to achieving these goals is to develop true F_1_ hybrids, often applying ER techniques to overcome various post-zygotic barriers. The objective is to provide valuable genetic materials for breeding strategies and to study the genetic effects of single chromosomes on plant traits, the number of genes that control a trait, their linkage relationships, and genetic improvement in Brassicaceae crops (reviewed by Kaneko and Bang [63]). For instance, alloplasmic lines were developed by exchanging cytoplasm through interspecific and intergeneric hybridizations followed by subsequent backcrosses with the donor species of the nuclear genome. These hybrids have acquired the advantages of CMS and useful agronomic traits [263].

Formanová et al. [133] reported the transfer of the cytoplasmic *pol* CMS and the nuclear gene *Rfp* from *B. napus* into a DH line of *Brassica rapa.* Embryogenesis from *Brassica rapa* microspores was obtained using sterile and fertile floral buds collected from BC_2_ generation plants [264]. Chromosome complement doubling was achieved by treating seedlings with a 0.34% (*w*/*v*) colchicine solution for 1.5 h. Cytogenetic analysis confirmed one of the DHs obtained a diploid characteristic of *B. rapa*. A cosmid library was generated from this plant. Therefore, Formanová et al. [133] generated a DH line of *B. rapa* containing the cytoplasm *pol* and its restorer *Rfp*. In addition, the authors formed a chromosome library from this line, from which it will be possible to clone the *Rfp* gene through genetic mapping approaches (Table 1 and Table 3).

Successful attempts to double the chromosome number, through the in vitro application of colchicine to F_1_ hybrids, were achieved in the cross between *Enarthrocarpus lyratus* (2*n* = 2*x* = 20, genome EnEn) and *B. oleracea* (2*n* = 2*x* = 20, genome CC) [131]. Specifically, 11 crosses between *E. lyratus* and *Brassica* spp. were performed. Ovary cultures at 10–15 DAP yielded hybrids only in two of the crosses, *E. lyratus* × *B. oleracea* and *E. lyratus* × *B. carinata* (2*n* = 34; amphidiploid). However, a higher number of hybrids was obtained by culturing pollinated ovaries at 4–6 DAP. Five crosses produced healthy plants: *E. lyratus* × *B. oleracea, E. lyratus* × *B. carinata, E. lyratus* × *B. campestris, E. lyratus* × *B. carinata,* and *B. juncea* × *E. lyratus* [131]. Nevertheless, an amphidiploid was obtained only from the hybrid *E. lyratus* × *B. oleracea* (2*n* = 38, genome EnEnCC). Cytological analysis showed 19 bivalents in most cells, and the plant showed 82% pollen fertility. About 20% of the amphidiploid flowers produced fruits and seeds when left to free pollination. Moreover, cytological studies of the hybrids demonstrated a partial homology between the genomes of *E. lyratus* and crop Brassicaceae. In addition, backcross progenies were obtained from several F_1_ hybrids (i.e., *E. lyratus* × *B. campestris*, *E. lyratus* × *B. oleracea*, *E. lyratus* × *B. napus*, *E. lyratus* × *B. carinata*, and *B. juncea* × *E. lyratus*) to develop male sterile alloplasmic lines [131].

Interspecific hybridization of *Brassica oleracea* ssp. *acephala* (2*n* = 2*x* = 18) × *Brassica rapa* ssp. *rapifera* (2*n* = 2*x* = 20) was performed. Furthermore, an in vitro placental pollination method was used to overcome certain stigma/style barriers [265]. *B. rapa* pollen was placed on opened *B. oleracea* ovaries with removed styles. Successful plantlet regeneration was achieved after in vitro culture of the hybrid embryos. The ER was performed with ovules isolated from the ovary at 14 DAP and transferred in vitro to a solidified MS medium [72] supplemented with 0.47 μM kinetin, 0.49 μM NAA, 10% (*v*/*v*) coconut water, and 2% sucrose. Moreover, chromosomes in the hybrid plants were doubled by dipping the roots for 24 h in 0.05% (*w*/*v*) colchicine solution. Cytometric analysis of nuclear DNA confirmed the hybrid nature of the plants. Furthermore, amphidiploid (2*n* = 4*x* = 38) *B. napus* plants were obtained [265] (Table 3).

In the genus *Brassica*, difficulties have often been encountered in obtaining well-developed embryos from interspecific hybridizations. For example, interspecific hybridization between *Brassica napus* and *B. rapa* resulted in a low hybridization rate [266]. A comparison between interspecific hybridization and self-fertilization materials demonstrated that egg cells’ genetic stability was greater than sperm cells. Therefore, an F_1_ with highly viable hybrid seed production can only be achieved through artificial pollination with other normal pollen. In this case, the authors adopted a technique developed some years ago, named DH inducer [267]. Fu et al. [267] obtained a synthetic octoploid rapeseed (2*n* = 8*x* ≈ 76, genome AAAACCCC). When this octoploid was used as the male parent in crosses with *Brassica oleracea* and *Brassica napus* [268], homozygous individuals like the female plants occurred in the progeny. The ploidy level typical of the female plants used in the crosses was confirmed by flow cytometry and SNP chip technology. Building upon these previous results, Zhou et al. [266] obtained interspecific maternal inbred offspring from egg cells of *B. napus* pollinated with the pollen of the DH inducer named Y3380. The FISH results confirmed that the obtained F_1_ was *B. napus* (2*n* = 4*x* = 38, genome AACC). Therefore, the DH inducer proved to be an innovative methodology for the interspecific hybridization of rapeseed. Potentially, other Brassicaceae species could benefit from the genetic resources of wild species in genetic improvement programs.

The irradiated pollen technique is the most successful haploidization technique within the Cucurbitaceae family [155]. In melon (*Cucumis melo*) (2*n* = 2*x* = 24), gamma (Co^60^γ-rays)- or soft X-ray-irradiated pollen induced in situ gynogenetic haploids through the parthenogenesis of egg cells [269,270,271,272,273]. In particular, Sauton and Dumas de Vaulx [269] applied ER to recover muskmelon haploid plants. They obtained haploid plants by the pollination of hermaphrodite flowers with irradiated (Co^60^ γ-rays) pollen and subsequently culturing ovules or immature embryos in vitro. In *Cucumis melo*, a classical method called “inspecting the seeds one by one” was used to find haploid embryos in the seeds of fruits pollinated with irradiated pollen [274].

Seedless citrus, one of the most desired quality traits, can be obtained through triploidy, as observed in many fruit species [275]. In citrus (*Citrus reticulata*) (2*n* = 2*x* = 18), triploid plants can be recovered by crossing diploid × diploid (e.g., *C. reticulata* × *C. sinensis* (2*n* = 2*x* = 18) and *Citrus clementine* (2*n* = 2*x* = 18) × *Citrus grandis* (2*n* = 2*x* = 18)), as well as diploid  ×  tetraploid ((e.g., *C. clementina* × (*C. paradise* × *C. tangerina*)) and tetraploid  ×  diploid combinations. The promotion of hybrid embryo growth is always supported by in vitro ER techniques [276,277,278,279]. Some examples are reported in Table 3. The influence of parents and environmental conditions on obtaining triploid hybrids was obvious. The genotype of the female parent had the most pronounced impact, as demonstrated by Aleza et al. [278], while a substantial interaction was observed between the male parent genotype and environmental conditions. Flow cytometer screening confirmed the hybrid nature of the obtained plantlets [278,279]. Furthermore, an effective method for obtaining triploid hybrids may be to pollinate non-apomictic tetraploid cultivars with pollen from diploid varieties. However, non-apomictic tetraploid lines were not found in the citrus germplasm [280]. Therefore, Aleza et al. [280] optimized a methodology based on in vitro shoot-tip grafting combined with treating the micro-grafted shoot tip with colchicine to achieve chromosome doubling and an efficient dechimerization strategy. Stable tetraploid plants were successfully obtained from different varieties. Tetraploid × diploid crosses were performed to recover triploid hybrids plants [280] (Table 3).

Singh and Nelson [281] developed methods to produce F_1_, amphidiploid, BC_1_, BC_2_, and BC_3_ fertile soybean plants from crosses between soybean plants (*Glycine max*, 2*n* = 40) with six accessions of *G. tomentella* (2*n* = 78) and one accession each of *G. tomentella*, *G. argyrea*, and *G. latifolia* (2*n* = 40). They also performed a methodology to create alloplasmic soybean plants by the introgression of cytoplasmic and valuable genetic traits (e.g., resistance to soybean rust) from *G. tomentella* into cultivated *G. max* (2*n* = 40) used as the male parent [281]. Immature seeds (19–21 DAP) from this cross were cultured in vitro to produce F_1_ plants (2*n* = 59). The fully developed F_1_ plants were perennials such as *G. tomentella*, and despite profuse blooming, they were sterile. The amphidiploid (2*n* = 118) plants were obtained through colchicine treatment, but they were dwarf and rarely produced one or two mature pods with seeds (Table 3). However, cutting from the amphidiploid plants showed normal morphological features, including leaves and flowers, and exhibited perennial growth. Moreover, the amphidiploid plants produced seeds after backcrossing with soybean. The BC_1_ plants (2*n* = 79) were further crossed with cultivated soybean. The chromosome numbers in BC_2_F_1_ plants ranged from 2*n* = 41 to 50. In BC_2_F_2_ to BC_3_F_1_ plants, the chromosome numbers ranged from 2*n* = 40 to 45. Fertile lines from these crosses were grown in the field [281]. This work represents the first report of the successful development of new alloplasmic soybean lines with *G. tomentella* cytoplasm.

#### 4.2.2. Monocot Species

Polyploidization is an attractive target in *Lilium* breeding programs, also to recover fertility in sterile F_1_ hybrids gained through interspecific crosses [282]. Polyploid plants, especially triploids and tetraploids, are highly valued in breeding programs due to their desirable traits, such as better flower quality, robust stems, and thicker and larger flowers [55,83]. Different chemicals used for mitotic chromosome doubling are colchicine, oryzalin, and surflan [283,284]. The choice of these substances is highly correlated with species. Ming et al. [285] reported that the highest percentage of tetraploid plants was produced in *L. pumilum*, *L. sargentiae*, and *L. tsingtauense* when treated with 0.04% colchicine for 24 h or 0.02 and 0.04% colchicine for 48 h. Triploid progenies of *Lilium* could be produced by crossing Oriental × Asiatic hybrids [286]. Crossing *Lilium auratum* (2*n* = 2*x* = 24) with *L. henryi* (2*n* = 2*x* = 24) resulted in F_1_ hybrids producing a significant frequency of unreduced gametes (2*n*) due to incomplete first meiotic division (first division restitution (FDR)) [287]. These F_1_ plants were successfully used for crosses with hybrids of Oriental *Lilium* varieties, resulting in triploid plants. This material holds potential value because recombinant chromosome segments can be introgressed into further generations in breeding programs [287] (Table 4).

Barba-Gonzalez et al. [288] also successfully used the production of sexual polyploid lily cultivars through unreduced gametes.

Li et al. [289] reported the development of triploids (2*n* = 3*x* = 33) in day lily (*Hemerocallis*) germplasm by crossing diploid (2*n* = 2*x* = 22) cultivars with tetraploid (2*n* = 4*x* = 44) cultivars. Analogously, Yao and Cohen [290] obtained triploid (2*n* = 3*x* = 48) hybrid plants from crosses between diploid (2*n* = 2*x* = 32) *Zantedeschia elliottiana* and tetraploid (2*n* = 4*x* = 64) cultivars of the winter-dormant section in the genus *Zantedeschia*. The production of triploid hybrids in day lilies and *Zantedeschia* is primarily aimed at achieving sterility, which prevents fertilization and prolongs the lifespan of the flower, making it desirable in the flower market (Table 4).

*A. officinalis* L. is a diploid species (2*n* = 2*x* = 20), but within the *Asparagus* genus, several wild species exhibit different ploidy levels: *A. prostratus*, *A. acutifolius*, *A. maritimus*, and some landraces of *A. officinalis* are tetraploids (4*x*); *A. maritimus* is hexaploid (6*x*); and *A. macrorrhizus* is dodecaploid (12*x*) [291,292]. The hybridization of genotypes with different ploidy levels in asparagus [293,294,295,296], as well as the induction of autotetraploid, auto-octoploid, or triploid genotypes [297,298,299], have occasionally resulted in plants with valuable agronomical traits. However, different ploidy levels also involve diverse challenges in asparagus breeding, such as crossing incompatibility, overcoming by ER, and genetic instability [299,300,301].

Hybrids between *Ae. ovata* (2*n* = 4*x* = 28) and *S. cereale* (2*n* = 2*x* = 14) were produced (2*n* = 3*x* = 21) [302]. Before crossing, the stigmas of *Ae*. *ovata*, used as female parents, were treated with GA_3_. The proper development of the embryo was supported by the use of ER. Differentiated embryos were placed on medium B5 [71] without PGRs, while the undifferentiated ones were placed on B5 supplemented with 2 mg L^−1^ 2,4-D for callus induction. Only four hybrid plants were obtained from a single cross by culturing differentiated embryos. These hybrids were initially sterile but regained fertility through chromosome doubling via colchicine treatment. The resulting amphidiploids (2*n* = 6*x* = 42) displayed morphological similarities to the hybrid plants but had lower tillering ability, broad leaves, and larger spikes [302].

Doubled haploid (DH) technology is a valuable approach for accelerating genetic gain through a shortened breeding cycle. In wheat and barley, DHs obtained from immature microspores or in vitro ER following interspecific hybridization are important strategies for breeding and generating populations for gene mapping purposes (reviewed by Broughton et al. [303]; Patial et al. [304]).

Initially, interspecific hybridization was the primary method to obtain barley haploids [305]. Kasha and Kao [306] reported haploid production through crosses of *H. vulgare* with *H. bulbosum*, while Fedak [307,308] obtained haploids from crosses of barley × rye. The method provides crossing *Hordeum vulgare* (2*n* = 2*x* = 14) as the female parent, with diploid *H. bulbosum* (2*n* = 2*x* =14) as the male parent. The resulting hybrid embryos contain chromosomes from both parents, but the chromosomes of *H. bulbosum* are progressively eliminated during embryogenesis. As a result, the embryos become haploid with the chromosome complement of *H. vulgare* (*n* = 7). The progressive elimination of *H. bulbosum* chromosomes also occurs in the endosperm, which prevents haploid embryo development. Therefore, haploid embryos must be rescued at 12–14 DAP by growing them in vitro before the degeneration of the endosperm [304]. Sanei et al. [309] demonstrated that the loss of centromeric histone H3 (CENH3) from centromeres preceded uniparental chromosome deletion in interspecific barley hybrids (reviewed by Britt and Kuppu [310]). This observation has led to the potential use of CENH3 modification for haploid production. Recently, Wang et al. [311] introduced a methodology involving the crossing of a female maize line heterozygous for the *cenh3* null mutation (*cenh3*/*CENH3*) with wild-type pollen. This technique resulted in approximately 5% of seeds containing haploid paternal embryos. In wheat, the hexaploid nature poses challenges in genetic manipulation to obtain haploids due to the presence of 12 CENH3 alleles. However, Lv et al. [312] identified, among the transformed plants, a specific type of mutation in *CENH3α-A*, named *Restored FrameShift* (*RFS*). This mutation, along with knockout alleles for *CENH3α-B* and *CENH3α-D*, activated haploid induction in wheat (reviewed by Widiez [313]).

Another method for obtaining haploids in barley became established with the discovery of the *hap* mutant [314]. The *hap* gene prevents egg cell fertilization, while the polar nucleus is fertilized, and usually, the endosperm develops. The egg cell is stimulated to grow and produce a haploid embryo. Therefore, if a plant homozygous for the *hap* gene is used as a female parent in crosses with other cultivars, a portion of the progeny (approximately 8%) results in maternal haploids. Currently, reciprocal crosses have not produced haploids. This methodology is attractive because it does not use in vitro culture but is quite complex in the hybridization procedures [304]. Numerous endogenous and exogenous factors influence haploid production in barley using both in vitro and in vivo methods (reviewed by Patial et al. [304]). Genetic factors, culture media composition, pollen development stage, ovule culture stage in interspecific hybridization, physical or chemical pre-culture treatments of anthers and microspores, and the occurrence of albino plants all influence haploid production in barley. Spontaneous duplication of the haploid chromosome complement leads to a high incidence of DHs in barley (60–80%). However, spontaneous induction of DHs is often very low, necessitating the use of polyploidization substances, such as colchicines, nitrous oxide, caffeine, and herbicides (e.g., amiprophos-methyl, oryzalin, trifluralin, and pronamide).

Uniparental chromosome deletion is a process not restricted to the *H. vulgare* × *H. bulbosum* cross but common in interspecific and intergeneric cereal hybrids. Polgári et al. [315] characterized a population of 218 independent plants derived from 2 crosses of *Triticum aestivum* (2*n* = 6*x* = 42, female) × *H. vulgare* (male). At 1 DAP, spikes were treated injecting 100 ppm 2,4-D to promote pseudo seed development. Germination and regeneration of embryos taken at 14–16 DAP occurred on a modified N6 medium [73,81]. The analysis showed a nearly identical frequency of haploids and complete hybrids (20.5% and 19.5%, respectively). The remaining plants were hypolyploids (partial hybrids) with no preference for the elimination of individual barley chromosomes in a wheat background.

The culture of embryos obtained by hybridization between *T. aestivum* and *Zea mays* (2*n* = 2*x* = 20) is a widely used method for obtaining haploid and, later, homozygous lines (DH) of wheat [316,317,318,319]. This technique is quite laborious and involves several steps: (i) Emasculation of the wheat flower and pollination of the emasculated flower with maize pollen. (ii) Hormone treatments of the flowers with growth regulators—Deveax [318] applied auxin to each wheat floret at 1 DAP. (iii) ER. (iv) Regeneration of the haploid shoot in culture medium (during this stage, maize chromosomes were eliminated). (v) Chromosome doubling [320]. The authors observed that DH production efficiency mainly depended on the maize and wheat genotypes used in the crossing, the health status of the plants (e.g., the absence of biotic and abiotic stresses), and their excellent development.

Despite technical difficulties, the induction of haploids in hexaploid wheat by hybridization with maize has long been widely documented [321,322,323,324,325]. Additionally, although more rarely, haploids have also been induced in *Triticum aestivum* by hybridization with pearl millet (*Pennisetum glaucum*, 2*n* = 2*x* = 14) [326] and with *Imperata cylindrical* (2*n* = 2*x* = 20) [327,328]. Wheat DHs were also obtained by pollination with *Coix lachryma-jobi* (2*n* = 2*x* = 20) [329] (Table 4).

To obtain haploids in *Triticum turgidum* ssp. *turgidum* (2*n* = 4*x* =28), O’Donoughue and Bennett [330] crossed durum wheat (used as the female parent) with maize (used as the pollinator). The haploid seedlings of durum wheat were obtained after in vivo treatment of the ovules with 2,4-D for 2 weeks and subsequent in vitro culture of the embryos. To obtain seedlings from the durum wheat cv. “Wakona”, in addition to treatment with 2,4-D, it was essential to add silver nitrate (AgNO_3_), an ethylene inhibitor. In durum wheat, this procedure yielded 1.7 to 3.3% haploid plants. The genotype used as the female plant significantly influenced the percentage of haploids produced [330].

Doğramaci-Altuntepe and Jauhar [331] studied the effect of single chromosomes of durum wheat on crossability with maize by cytologically characterizing the recovered haploids. With this objective, fourteen disomic D genome substitution lines of “Langdon” (LDN), a *Ph* mutant of LDN (*Ph1b*/*ph1b*), and normal “Langdon”, were used as female plants and crossed with maize. Notably, the disomic D genome substitution (LDN) lines are the major aneuploid stocks and have been widely used in genetic studies [332]. After pollinating wheat, the flowers were treated with auxin daily for two weeks. Out of the 55,358 pollinated flowers, 895 embryos were obtained, but only 14 germinated and developed into plants. The 5D(5B) substitution line proved to be the most satisfactory for embryo differentiation and haploid production, indicating that replacing 5D with 5B enhances the ability of durum wheat to produce haploids (*n* = 14). Fluorescent genomic in situ hybridization (GISH) analyses showed that the genome of the substitution haploids was composed of seven chromosomes from genome A, six chromosomes from genome B, and one chromosome from genome D [331].

The induction of haploids in durum wheat by crossing with maize is known [333,334,335,336]. To efficiently produce haploid plants, it is necessary to evaluate not only the genotype of the female plant but also the best pollinators. For this purpose, the influence of the male parent was evaluated in the production of gynogenetic embryos and haploid plants in *Triticum turgidum* crossed with maize and *Pennisetum glaucum* (2*n* = 2*x* = 14). Pollen mixtures of both maize and pearl millet were also evaluated in this work. The results showed no differences in embryo and haploid plant production among the four maize pollen samples. In contrast, significant genotypic differences were found for haploid production among the three pearl millet genotypes. Pollen mixtures yielded unsatisfactory results [336] (Table 4).

DHs can also be produced in oats (*Avena sativa*, 2*n* = 6*x* = 42) by pollinating oat flowers with maize pollen. Like other intergeneric or interspecific crosses during embryo development, the paternal parent’s (maize) chromosomes are eliminated, resulting in a haploid set of oat chromosomes in this cross. These embryos need to be cultured in vitro using ER. Chromosome doubling through colchicine treatment results in homozygous oat plants [31,337,338]. Warchoł et al. [339] conducted intensive hybridization work (*A. sativa × Z. mays*) using oats as a female parent to evaluate the effect of auxin treatments of oat flowers on the production of embryos, haploids, and DHs. The genotype significantly influenced embryo production, with the highest number of embryos per emasculated flower after treatment with dicamba or after treatment with 2,4-D, depending on the genotype considered. However, no genotypic influence on haploid plant development was observed, but an appreciable genotypic influence on DH production was found. In total, Warchoł et al. [339] obtained 149 haploid plants. A total of 61 plants survived the chromosome doubling procedure, and 52 fertile plants produced more than 5000 seeds. Skrzypek et al. [340] showed that in the oat × maize cross, light intensity also influenced haploid oat production. A light intensity of 110 μmol m^−2^ s^−1^ during in vitro embryo culture resulted in a higher percentage of embryo germination (38.9%), plant conversion (36.4%), and DH line production (9.2%) compared to lower light intensities. The developmental stage of the embryo and the culture medium were identified as crucial factors to obtain haploids and DH lines in oat × maize crossbreeding. Noga et al. [341] analyzed the germination behavior of embryos taken at different stages of development and determined the best combinations and concentrations of PGRs to use in the culture media for oats × maize crosses. The results showed that haploid embryos < 0.5 mm were unable to germinate, while embryos ≥ 1.5 mm exhibited successful germination. In addition, most haploid embryos germinated on a medium with 0.5 mg L^−1^ of NAA and 0.5 mg L^−1^ of kinetin, while other PGRs and/or different concentrations had fewer positive effects.

Kynast et al. [30] performed oat × *Zea mays* (2*n* = 2*x* = 20) crosses, and they were able to transfer maize chromosomes into oat DH to constitute monosomic and disomic addition lines. In particular, from in vitro rescue culture of more than 4000 immature F_1_ embryos, they recovered 379 F_1_ plants. Some of these plants were chimeric, meaning that different tillers from the same plant had different chromosome compositions. Nevertheless, meiotic restitution facilitated the development of unreduced gametes, and the self-pollination of these plants resulted in disomic addition lines (2*n* = 6*x* + 2 = 44) for maize chromosomes 1, 2, 3, 4, 6, 7, and 9. Monosomic addition lines (2*n* = 6*x* + 1 = 43) and haploid complement of oat (*n* = 3*x* + 1 = 22) were also recovered [30]. It is worth noting that the haploid embryos were produced using ER techniques, which involved doubling their chromosome numbers with colchicine to obtain DH oat plants. The oat haploids and partial hybrids with 1–3 maize chromosomes exhibited partial fertility. Self-fertilization of these partial hybrids resulted in DH oat plants with an added single maize chromosome (monosomic addition line) or an added pair of homologous maize chromosomes (disomic addition line) [30] (reviewed by Kynast et al. [342]; Davies and Sidhu [337]).

Skrzypek et al. [338] conducted a cytogenetic–molecular characterization of a large population obtained from the oat × maize cross, which included addition and DH lines. They analyzed over 130 oat lines obtained from the crosses to determine the presence of maize chromosomes. All plants analyzed possessed the full complement of oat chromosomes. PCR analysis facilitated the identification of maize chromosomes due to the presence of a *Ty3* retrotransposon (*Grande 1*) in the maize genome used for the cross. GISH analysis further revealed that eight oat × maize addition lines (OMAs) possessed varying numbers of complete maize chromosomes, ranging from 1 to 4. Some lines analyzed did not possess complete maize chromosomes but showed evidence of introgression of maize chromosome regions into oat chromosomes. A total of 27 OMA lines and 63 DH lines were fertile and produced seeds. Therefore, Skrzypek et al. [338] demonstrated that hybridization between oats and maize resulted in complete DH line generation and the incomplete deletion of maize chromosomes (OMA line generation).

One of the most important difficulties in oat DH production is the germination of haploid embryos obtained from *A. sativa* × *Z. mays* crosses. To analyze which factors influenced this aspect, Warchoł et al. [343] investigated the influence of culture medium and genotype on the germination of haploid embryos. Haploid embryos at 21 DAP were cultured on MS medium [72] with 3% sucrose, used as a control, and on a medium called 190-2 [344] supplemented with 6 or 9% maltose. All 22 tested genotypes produced haploid embryos, although no DH lines were obtained from the two varieties. The DC09002 oat genotype yielded the highest number of haploid embryos and DH lines. The most satisfactory germination results (about 7%) were achieved when embryos were cultured on 190-2 medium compared to MS medium (about 3% germination). Regarding the supplementation of the carbon source, the best outcomes in terms of haploid embryo germination and DH line production were obtained with 190-2 medium supplemented with 9% maltose and adjusted to pH 6.0.

To gain further insights into the poor germination of oat haploid embryos, Dziurka et al. [345] analyzed the anatomical structure and hormonal profile of haploid and zygotic embryos from two oat cultivars. Morphological analysis showed that haploid embryos at 21 DAP were smaller and less advanced in structure compared to zygotic embryos. Additionally, there were significant differences in the hormonal profiles between haploid and zygotic embryos. Haploid embryos exhibited lower levels of IAA and higher levels of cytokinins than zygotic embryos. The gibberellin (GA) levels varied in the two oat varieties examined. The importance of IAA, cytokinin, and GA levels in the correct development of the embryo is well established [346,347,348]. The observed differences in hormonal profiles between haploid and zygotic embryos may partially explain the slower development of haploid embryos. In addition, the elevated levels of jasmonic acid, salicylic acid, and ABA in haploid embryos also suggest that the low germination rate of haploid embryos might be dependent on the higher production of reactive oxygen species (ROS) compared to zygotic embryos [345].

In *S. cereale × Z. mays* cross, Marcińska et al. [349] evaluated the genotypic effect of *S. cereale* on haploid embryo production using fifteen winter rye genotypes. The authors also assessed how ovary enlargement and haploid embryo production were affected by synthetic auxin growth regulators, such as 2,4-D, dicamba, and picloram. Finally, the time between flower emasculation and pollination was analyzed. Although all factors analyzed significantly affected rye ovary enlargement, the frequency of haploid embryos produced was dependent only on the rye genotype. Twenty-one haploid embryos were obtained from six rye genotypes. The best stimulation of ovary enlargement was detected using dicamba compared with other synthetic auxins. The highest frequency of haploid embryos was obtained by pollinations performed six days after rye flower emasculation.

In rice breeding, the development of new varieties is one of the main goals, and DH technology plays a vital role in achieving this. In particular, DH technology is considered helpful for generating inbred lines from rice hybrids in a single generation and mapping quantitative trait loci (QTL) [350,351,352]. QTL analysis in DH progenies allowed the identification of genes involved in resistance to biotic (e.g., *Cnaphalocrocis medinalis*, Guenée [353], *Magnaporthe grisea* [354], and *Xanthomonas oryzae* pv. *oryzae* [355]) and abiotic stresses (e.g., lodging resistance [356] and drought tolerance [357]), as well as agronomical traits (e.g., internode length [358], biofortification [359], heading date, plant height [360], tiller number [361], grain quality, and spikelet fertility [362]).

Therefore, androgenesis has excellent potential for DH production, although its success heavily relies on the genotype. *Oryza glaberrima,* 2*n* = 2*x* = 24, responds better to anther culture than *Oryza sativa*, 2*n* = 2*x* = 24, and the *japonica* sub-group is more responsive to androgenesis than *indica* rice [363,364]. The success of androgenesis is controlled by many factors, including genotype, stage of pollen grain development, pre-treatment conditions, choice of the most suitable basal media, concentrations of PGRs and carbon sources, and environmental conditions (e.g., photoperiod, light intensity, and temperature) during growth in vitro. In particular, Ali et al. [365] demonstrated that genotype and media significantly influence callus induction frequency and green plantlet regeneration efficiency. Nevertheless, callus induction from anther culture has been induced in many rice genotypes [366,367]. Mainly, two basal culture media, MS [72] and N6 [73], have been used for callus regeneration [368,369,370]. Various protocols suggesting different combinations of PGRs (BAP, IAA, NAA, and kinetin) have been applied to regenerate androgenetic shoots [366,370,371,372,373]. In the *japonica* cultivar, Ferreres et al. [374] demonstrated that adding 150 mg L^−1^ colchicine to the induction medium increased regeneration frequency and DH production.

Lantos et al. [375] analyzed the genotypic effect on rice androgenesis by culturing in vitro anthers of five genotypes. The authors also assessed the effect of various PGRs and culture media combinations on callus production and the regeneration of green and albino seedlings. For induction of calli, the N6 medium [73] with two different combinations of PGRs (first: 2.5 mg L^−1^ NAA, 1 mg L^−1^ 2,4-D, and 0.5 mg L^−1^ kinetin; second: 2 mg L^−1^ 2,4-D and 0.5 mg L^−1^ BAP) was used. For regeneration, two media (MS [72] and N6 [73]) and two combinations of PGRs (first: 1 mg L^−1^ NAA, 1 mg L^−1^ BAP, and 1 mg L^−1^ kinetin; second: 1.5 mg L^−1^ BAP, 0.5 mg L^−1^ kinetin, and 0.5 mg L^−1^ NAA) were compared. The highest production of green plant was achieved using the N6 induction medium, supplemented with 2.5 mg L^−1^ NAA, 1 mg L^−1^ 2,4-D, and 0.5 mg L^−1^ kinetin, and the MS regeneration medium, supplemented with 1 mg L^−1^ NAA, 1 mg L^−1^ BAP, and 1 mg L^−1^ kinetin. Among the genotypes tested, one responded with the highest production of green seedlings (95.2 green seedlings/100 anthers) compared to the other. Flow cytometric analysis identified the origin of haploid callus from microspores, and a high rate of spontaneous doubling of the chromosomal complement was observed, ranging from 38.1 to 57.9%, depending on the genotype. Additionally, DH lines were selected in field experiments to evaluate the morphological characteristics of agronomic interest. More recently, Lantos et al. [376], in an attempt to improve the culture media using other genotypes of *indica* rice obtained through androgenesis, successfully produced 48 haploid, 55 diploid, 2 tetraploid, and 1 myxoploid seedlings. The production of haploids remained greatly conditioned by the genotype used for the anther culture, and a high rate of spontaneous chromosome doubling (51.89%) facilitated the production of DH plants.

In an extensive work on *indica* rice, Dash et al. [377] analyzed the in vitro behavior of 6 genotypes using 12 different culture media and 5 different cold treatments of anthers (i.e., stress time). The most effective methodology for callogenesis in all genotypes tested was N6 medium [73] supplemented with 2.0 mg L^−1^ 2,4-D, and 0.5 mg L^−1^ BAP, along with 7 days cold pre-treatment. However, Pattnaik et al. [378] found that the pre-treatment of spikes at 10 °C for 2 days was most effective for both callus formation and regeneration. It appears that the optimal cold pre-treatment may depend on the genotype. For genotypes of Mediterranean *japonica* rice, a 9-day cold pre-treatment at 5.0 °C was optimal for inducing a higher rate of anther-derived callus [379]. Regarding green shoots, Dash et al. [377] demonstrated that the most efficient culture medium was the basal MS medium [72] supplemented with 0.5 mg L^−1^ NAA, 2.0 mg L^−1^ BAP, and 1.0 mg L^−1^ kinetin. Notably, the highest regeneration rate (58.25%) was observed in one genotype. This suggested that although this culture medium was efficient for all hybrid varieties tested, a significant genotype influence on both callogenesis and regeneration was effective. Interestingly, the addition of 5 g L^−1^ proline to the best culture medium increased the regeneration rate to 85.99%. Rooting the regenerated shoots was a challenge among various genotypes. MS basal medium supplemented with 1.0 mg L^−1^ NAA, 0.1 mg L^−1^ kinetin, and 50 g L^−1^ sucrose was the most efficient for adventitious root induction in all hybrid varieties tested. This study highlighted a regeneration protocol that is relatively independent of the *indica* rice genotype used. The authors envisaged its use to produce DHs required in genetic improvement programs for *indica* rice [377].

Transposon mutagenesis involves the use of transposable genetic elements that integrate into a recipient genome, generating random insertion mutations easily identified. Tai et al. [380] developed an inducible *COKC* transposon by combining the transposase gene with a chemically inducible *PR-1a* promoter. *COKC* was introduced into rice plants and activated by salicylic acid (SA) to trigger transposition events. The assumption was that transposition could be induced during anther culture, allowing selected mutations to be transmitted and recovered in the homozygous state in DH regenerated plants [381]. Yang and Charng [381] used five independent transgenic lines, each containing a single copy of *COKC*, as the plants from which anthers were taken for callus production and regeneration of DH plants. The authors observed that inducible transposon was active during callus regeneration, resulting in around 5% of the mutants possessing transposon events in the homozygous condition. Consequently, SA-induced transposition in male rice gametes, followed by anther culture, produced independent DH mutants. This system demonstrated significant applicability for functional genomic studies and induction genetic variability in rice improvement programs.

#### 4.2.3. Genome Editing to Induce Haploid Plants

Clustered Regularly Interspaced Short Palindromic Repeats (CRISPR)-based genome editing can also be a methodology for obtaining haploids [382]. In wheat, a haploid induction rate of 18.9% was found in *MATRILINEAL* (*MTL*)-edited T_1_ plants using the CRISPR/SpCas9 system [383]. *MTL*, also named *NOT LIKE DAD* (*NLD*) and *PHOSPHOLIPASE A1* (*PLA1*), encodes for a pollen-specific phospholipase A1 that can trigger haploid induction in maize as well [384]. The knockout of *OsMATL* in rice can induce up to 6% haploid embryos [385]. The same method has documented similar results in foxtail (*Setaria italica*, 2*n* = 2*x* = 18) [386]. Recently, a system was developed to produce gynogenetic haploid plants by the mutagenesis of genes encoding egg-cell-specific aspartic endopeptidases (ECSs) [387]. Zhangh et al. [387] demonstrated that ESCs play a crucial role in ensuring the fusion of the male and female nucleus after fertilization. By using the CRISPR/Cas9 system, the *Osecs1*/*Osecs2* double mutant induced haploids in rice. Therefore, this methodology has shown great potential for genetic improvement in rice, especially in androgenesis recalcitrant genotypes.

By knocking out of orthologs of *PLA1/MATL/NLD* or *DOMAIN OF UNKNOWN FUNCTION 679 membrane protein* (*DMP*), in vivo haploid induction has been proven as feasible also in dicot species, such as *Solanum lycopersicum* [388], *Nicotiana tabacum* [389], *Brassica napus* [390], *Medicago truncatula* [391], *Brassica oleracea* [268,392], and *Solanum tuberosum* [393].

One challenge in using genome editing to obtain haploids is identifying haploid embryos (HID), which is essential before chromosome complement doubling to obtain DH plants [394]. *R1-navajo* (*R1-nj*) is one of the most widely used markers for HID in monocots in relation to clear pigmentation in both scutellum and aleurone [395]. For other species, including dicot and monocot species with high oil concentration in the seeds, other markers have been identified to enable accurate HID [396]. The observation that betalains, pigments derived from tyrosine, constitute one of the main pigments in plants of the order Caryophyllales [397] suggested the genes coding for the metabolic enzymes of these pigments as reporter genes for HID. Betalain biosynthesis requires three genes: *CYP76AD1*, *BvDODA1,* and *cDOPA5GT* [398]. The open reading frame (ORF) containing the three genes, called *RUBY*, has been used as a reporter gene to monitor gene expression in transgenic plants [399]. Wang et al. [396] also demonstrated that the *RUBY* gene enables HID in maize and tomato. In maize, the results of Wang et al. [396] showed that haploid embryos of maize at 10 DAP recovered in vitro, accumulated a significant amount of betalains. In tomato, the authors observed that the *RUBY* gene induced deep-red pigmentation in the rootlets of the haploids, enabling easy and accurate identification. Cytological analysis confirmed 100% correct haploid identification in both immature maize embryos and tomato haploid embryos. Therefore, the *RUBY* reporter gene was a useful system for the haploid selection of embryos in both monocot and dicot species.

Although genome editing is a methodology for haploid induction in planta, in some cases, it requires the ER of immature embryos. In some species (e.g., maize and potato), germination of haploid embryos requires in vitro culture to facilitate their full development or in vitro treatments with GA to break seed dormancy.

**Table 3 plants-12-03106-t003:** Some examples where embryo rescue (ER) supports ploidy level changes in dicot crops.

Crops	Methods	Ploidy Level	Main Objective	References
*Cichorium intybus* (2*n* = 2*x* = 18)	^1^ IG cross with *Cicerbita alpina*	Gynogenetic haploids	^2^ DHs	[242]
*Lactuca sativa* (2*n* = 2*x* = 18)	IG crosses with *Helianthus* spp.	Gynogenetic haploids	DHs	[243]
*Rosa × hybrida* (2*n* = 4*x* = 28)	^3^ IS crosses with *Rosa* spp. (2*n* = 4*x*, 5*x*, 6*x*)	Triploids and tetraploids	Sterility and variability	[244]
*Primula sinensis* (2*n* = 2*x* = 24) (male)	IS cross with *P. filchnerae* (diploid)	Triploids and hexaploids	Genetic variability	[246]
*Selenicereus megalanthus* (2*n* = 4*x* = 44)	Gynogenesis	DHs, triploids, tetraploids	Cacti breeding	[247]
*Hylocereus megalanthus* (2*n* = 4*x* = 44) (male)	IS crosses with *H. megalanthus* and *H. undatus* (diploids)	Diploids, triploids, tetraploids	Cacti breeding	[248]
*Gentiana* spp. (2*n* = 2*x* = 26)	Gynogenesis	Haploids and diploids	DHs	[249]
*Begonia semperflorens* (2*n* = 2*x*) and (2*n* = 4*x*)	IS cross with *B.* “Orange Rubra” (2*n* = 2*x*)	Haploids, DHs, and others	*Begonia* spp. breeding	[251]
*Dianthus caryophyllus* (2*n* = 2*x* = 30)	Impollination with RX-treated pollen and gynogenesis	Haploids and DHs	Carnation breeding	[252]
*Cyclamen persicum* (2*n* = 2*x* = 48)	IS cross with *C. hederifolium* (2*n* = 2*x* = 34)	IS sterile hybrids	Fertile ^4^AD (2*n* = 82)	[255]
*Cyclamen persicum* (2*n* = 2*x* = 48)	IS cross with *C. purpurascens* (2*n* = 2*x* = 34)	IS sterile hybrids	Fertile AD	[257]
*Cyclamen persicum* (2*n* = 2*x* = 48)	IS cross with *C. graecum* (2*n* = 4*x* = 84)	IS low fertile hybrids	Cyclamen breeding	[256]
*Cyclamen persicum* (2*n* = 4*x* = 96)	IS cross with *C. graecum* (2*n* = 4*x* = 84)	IS fertile hybrids	Disease resistance	[256]
*Jatropha curcas* (2*n* = 2*x* = 22)	IG cross with *Ricinus communis* (2*n* = 2*x* = 20)	IG sterile hybrids	AD	[258]
*Vitis vinifera* × *V. labrusca* (2*n* = 4*x* = 78)	IG cross with *V. vinifera* diploid (2*n* = 2*x* = 38)	Haploids and triploids	DHs	[260]
*Enarthrocarpus lyratus* (2*n* = 2*x* = 20)	IG cross with *B. oleracea* (2*n* = 2*x* = 18)	IG sterile hybrids	Fertile AD (2*n* = 38)	[131]
*Brassica rapa* (2*n* = 2*x* = 16)	Androgenesis from ^5^ BC_2_ of *B. rapa* × *B. napus*	Haploids and DHs	*Rfp* gene cloning	[133]
*Brassica oleracea* ssp. *acephala* (2*n* = 2*x* = 18)	IS cross with *B. rapa* ssp. *rapifera* (2*n* = 2*x* = 20)	IS hybrids	Fertile AD (2*n* = 38)	[265]
*Cucumis melo* (2*n* = 2*x* = 24)	Irradiated pollen (Co^60^ γ- and X-rays)	Gynogenetic haploids	DHs	[269,270,271,272,273]
*Citrus reticulata* (2*n* = 2*x* = 18)	IS cross with *C. sinensis* (2*n* = 2*x* = 18)	Sterile triploids	Seedless	[278,279]
*Citrus clementina* (2*n* = 2*x* = 18)	IS cross with *C. grandis* (2*n* = 2*x* = 18)	Sterile triploids	Seedless	[277]
*Citrus clementina* (2*n* = 2*x* = 18)	IS with *C. paradise* × *C. tangerine,* tetraploid	Sterile triploids	Seedless	[278]
*Glycine max* (2*n* = 2*x* = 40) (male)	IS with *G. tomentella* (2*n* = 78)	Sterile hybrids (2*n* = 59)	AD (2*n* = 118)	[281]
*Solanum lycopersicum* (2*n* = 2*x* = 24)	Genome editing of the *SlDMP* gene	Gynogenetic haploids	DHs	[388]
*Nicotiana tabacum* (2*n* = 4*x* = 48)	Genome editing of the *NtDMP* gene	Gynogenetic haploids	DHs	[389]
*Brassica napus* (2*n* = 4*x* = 38)	Genome editing of the *BnDMP* gene	Gynogenetic haploids	DHs	[390]
*Medicago truncatula* (2*n* = 2*x* = 16)	Genome editing of the *MtDMP* gene	Gynogenetic haploids	DHs	[391]
*Brassica oleracea* (2*n* = 2*x* = 20)	Genome editing of the *BoDMP* gene	Gynogenetic haploids	DHs	[392]
*Solanum tuberosum* (2*n* = 4*x* = 48)	Genome editing of the *StDMP* gene	Gynogenetic haploids	DHs	[393]

^1^ IG—intergeneric; ^2^ DHs—doubled haploids; ^3^ IS—interspecific; ^4^ AD—amphidiploid; ^5^ BC_2_—backcross 2.

**Table 4 plants-12-03106-t004:** Some examples where the embryo rescue (ER) supports ploidy level changes in monocot crops.

Crops	Methods	Ploidy Level	Main Objective	References
*Hemerocallis* (2*n* = 2*x* = 22)	^1^ IS cross with tetraploid cultivars	Triploids	Sterility	[289]
*Zantedeschia elliottiana* (2*n* = 2*x* = 32)	IS cross with tetraploid cultivars	Triploids	Sterility	[290]
*Lilium auratum* (2*n* = 2*x* = 24)	IS cross with *L. henryi* (2*n* = 2*x* = 24)	Hybrids producing 2*n* gametes	Polyploidy	[287]
*Lilium* spp. Orientalis (2*n* = 2*x* = 24)	IS cross with *Lilium* spp. (Asiatic hybrid)	Triploids	Variability	[286]
*Aegilops ovate* (2*n* = 4*x* = 28)	^2^ IG cross with *Secale cereale* (2*n* =2*x* = 14)	Sterile triploids (2*n* = 3*x* = 21)	Fertile ^3^ AD	[302]
*Avena sativa* (2*n* = 6*x* = 42)	IG cross with *Zea mays* (2*n* = 2*x* = 20) (male)	^4^ AL (2*n* = 6*x* = 42 +1, 2)	^5^ OMA	[30,338]
*Oryza sativa* (2*n* = 2*x* = 24)	Androgenesis (anther culture)	Haploids	^6^ DHs	[350,351,352,353,354,355,356,357,358,359,360,361,362,363,364,365,366,367,368,369,370,371,372,373,374,375,376,377,378,379]
*Hordeum vulgare* (2*n* = 2*x* = 14)	IS cross with *H. bulbosum* (2*n* = 2*x* =14) (male)	Gynogenetic haploids	DHs	[304,305,306,307,308]
*Hordeum vulgare* (2*n* = 2*x* = 14)	Cross with *H. vulgare* (genotype: *hap/hap*) (female)	Gynogenetic haploids	DHs	[304,314]
*Triticum aestivum* (2*n* = 6*x* = 42)	IG cross with *H. vulgare* (2*n* = 2*x* = 14) (male)	Gynogenetic haploids	DHs	[315]
*Triticum aestivum* (2*n* = 6*x* = 42)	IG cross with *Z. mays* (2*n* = 2*x* = 20) (male)	Gynogenetic haploids	DHs	[316,317,318,319,320,321,322,323,324,325]
*Triticum aestivum* (2*n* = 6*x* = 42)	IG cross with *Pennisetum glaucum* (2*n* = 2*x* = 14) (male)	Gynogenetic haploids	DHs	[326]
*Triticum aestivum* (2*n* = 6*x* = 42)	IG cross with *Imperata cylindrical* (2*n* = 2*x* = 20) (male)	Gynogenetic haploids	DHs	[327,328]
*Triticum aestivum* (2*n* = 6*x* = 42)	IG cross with *Coix lacryma-jobi* (2*n* = 2*x* = 20) (male)	Gynogenetic haploids	DHs	[329]
*Triticum turgidum* ssp. *turgidum* (2*n* = 4*x* =28)	IG cross with *Z. mays* (2*n* = 2*x* = 20) (male)	Gynogenetic haploids	DHs	[330]
*Triticum turgidum* (2*n* = 4*x* =28)	IG cross with *Z. mays* (2*n* = 2*x* = 20) (male)	Gynogenetic haploids	DHs	[333,334,335,336]
*Triticum turgidum* (2*n* = 4*x* =28)	IG cross with *P. glaucum* (2*n* = 2*x* = 14) (male)	Gynogenetic haploids	DHs	[336]
*Avena sativa* (2*n* = 6*x* = 42)	IG cross with *Z. mays* (2*n* = 2*x* = 20) (male)	Gynogenetic haploids	DHs	[31,337,338,339,340,341,343,345]
*Secale cereale* (2*n* = 2*x* = 14)	IG cross with *Z. mays* (2*n* = 2*x* = 20) (male)	Gynogenetic haploids	DHs	[349]
*Zea mays* (2*n* = 2*x* = 20) (male)	Cross with *Z. mays* (genotype: *cenh3/CENH3*)	Androgenetic haploids	DHs	[307]
*Triticum aestivum* (2*n* = 6*x* = 42)	Genome editing of the *MATRILINEAL* (*TaMTL*) gene	Gynogenetic haploids	DHs	[383]
*Zea mays* (2*n* = 2*x* = 20)	Genome editing of the *ZmMATL* gene	Gynogenetic haploids	DHs	[384]
*Oryza sativa* (2*n* = 2*x* =24)	Genome editing of the *OsMATL* gene	Gynogenetic haploids	DHs	[385]
*Oryza sativa* (2*n* = 2*x* =24)	Genome editing of the *OsECS1/ECS2* genes	Gynogenetic haploids	DHs	[387]
*Setaria italica* (2*n* = 2*x* = 18)	Genome editing of the *SiMATL* gene	Gynogenetic haploids	DHs	[386]

^1^ IS—interspecific; ^2^ IG—intergeneric; ^3^ AD—amphidiploid; ^4^ AL—addition line; ^5^ OMA—oat-maize addition line; ^6^ DHs—doubled haploids.

### 4.3. Analysis of Embryogenesis with Embryo Rescue Technical Holder

The in vitro culture of ovules a few hours after pollination [400] or isolated zygotes [401] provides an excellent system to study embryogenesis, as this process is more complex to analyze in plants. Therefore, in vitro investigations during embryo development have already been established in maize [402,403], barley [404], wheat [401,404,405,406,407,408], rice [409,410,411,412], tobacco [413], and *Setaria viridis* [414]. The use of histological and other microscopic tools is simplified during in vitro culture. This allowed detailed analysis of all stages of embryonic development, from the first zygotic division to the morphology of the mature embryo (Figure 1). Furthermore, transcriptomic and proteomic analyses on isolated egg cells, zygotes, and proembryos developing in vitro were conducted in wheat [415], maize [412,416,417,418], tobacco [419], and rice [412,420,421] to reveal genes and proteins that play crucial roles from the earliest stages of embryogenesis to full embryo development.

ER techniques have also proven to be helpful for studying embryonic development and molecular characterization of embryo-lethal mutants. In sunflower, the lethal mutant *non-dormant-1* (*nd-1*) (Figure 2A), characterized by extreme albinism, dies at the cotyledonary stage at light intensities above 100 μmol photons m^−2^ s^−1^ [422]. The *nd-1* embryo cultured in vitro at low light intensity (1 µmol m^–2^ s^–1^) developed pale-green cotyledons (Figure 2B) because *nd-1* cotyledons accumulated chlorophylls under such conditions. In very low light intensity, the pigments extracted from *nd-1* seedlings showed absorption maxima at 379, 400, and 426 nm (in petroleum ether; Figure 2C), suggesting the accumulation of *ζ-carotene* [423]. In fact, based on genomic Southern analysis, the *nd-1* genome disclosed a large deficiency at the *ζ−carotene desaturase* (*ZDS*) locus [423].

In carrots, the temperature-sensitive *ts11* mutant stopped its development early [424]. This developmental defect could be overcome by adding a mixture of proteins secreted by wild-type embryo cultures to the culture medium [424]. From this mixture of secreted proteins, a 32 kD glycoprotein, called extracellular protein 3 (EP3), was purified, which allowed the embryonic development of *ts11* to be completed in vitro at a non-permissive temperature [425]. EP3 was identified as a glycosylated acidic endochitinase. The addition of 32-kD endochitinase to in vitro cultures of mutant embryos at a non-permissive temperature appeared to promote the proper formation of an embryonic protoderm [425].

Breeding of *low-phytic-acid* (*lpa*) crops has been considered a potential way to increase the nutritional quality of crop products [426]. Eight independent *lpa* rice mutants were obtained from both *indica* and *japonica* subspecies, using physical and chemical mutagenesis. Among them, five were non-lethal, while the other three were homozygous lethal [427]. None of the lethal lines could produce homozygous *lpa* plants through seed germination. However, for two of them, the viability was recovered through in vitro culture of mature embryos. This result made it possible to analyze the development of the seedlings. During in vitro culture, the performance of mutant lines was significantly inferior to their wild-type parents. They showed a lower germination rate, more susceptibility to contamination, and slower growth than their sibling seeds, which contain standard phytic acid content [427]. Homozygous lethal *lpa* mutations were also reported in rice, where seeds homozygous for the *lpa* locus failed to germinate and develop plants [428]. A reduction in seed viability was also reported in *lpa* mutants of maize [429,430] and soybean [431].

In maize, three single-gene mutants, named *empty pericarp* (*emp*) based on their extreme reduction in endosperm development, already evident at 12 DAP, were characterized [432]. Their analysis was possible by in vitro culture of the immature embryos. Mutant embryos on a segregating ear were recognizable by their small size. Immature embryos were taken at 18, 24, and 36 DAP and subsequently cultured as described by Consonni et al. [433] on solidified MS basal medium [72]. After 20 days of culture, the germination percentage and seedling elongation were evaluated. Histological analysis revealed partial endosperm development and adhesion loss between the pedicel tissues and the basal transfer layer. In the endosperm, programmed cell death was delayed compared to the control. Although the mutant was not impaired in morphogenesis stages, the *emp* embryos showed apparent growth retardation.

The *NCS6* mutation in maize is a partial deletion of the mitochondrial *cytochrome oxidase subunit 2* (*COX2*) gene. The mutant can survive heteroplasmatically in the plant. ER can also recover homoplasmic embryos on a solid substrate. Embryos taken from the ears were cultured in vitro [434]. During the in vitro culture, slow-growing callus production was observed, which was shown to be homoplasmic for the *cox2* mutation. However, most of the rescued embryos were heteroplasmic for normal and mutant genes, and callus proliferation was more abundant. No differentiation of adventitious shoots was observed when homoplasmic *cox2* callus cultures were placed on the regeneration medium. In contrast, heteroplasmic calluses were able to regenerate under the same culture conditions. These studies suggested that the mitochondrial cytochrome oxidase role was not essential for callus proliferation. However, it was evident that *COX2* was crucial for acquiring cellular totipotency and developing regenerated shoots [434].

### 4.4. Embryo Rescue to Overcome Seed Dormancy, Fruit Drops; to Propagate Rare Plant Species; and to Speed up the Selection Processes

In vitro culture of zygotic embryos has proven very useful for overcoming seed dormancy (Figure 1). As is well known, the seeds of many plant species germinate only after a more or less prolonged period of dormancy due to physiological inhibitors, mainly hormonal (e.g., abscisic acid). By transferring immature embryos in vitro at a certain stage of development, which can vary depending on the species, the effect of the inhibitors can be suppressed, and the seeds can germinate rapidly [11,435,436,437].

The length of the seed-to-seed cycle is often a limiting factor in plant breeding for the development of recombinant inbred lines (RILs) required for genetic analysis and in molecular marker technology [438]. The rapid germination of immature embryos accelerates the selection process, reducing the time needed to complete a generation. For instance, in sunflower, the cycle from the germination of the immature embryo to seed production can be completed in 60–70 days, on average, allowing 4–5 generations per year in controlled environments [439,440]. Moreover, breeding cycles were shortened by ER also in rose [441], lily [442], lentil [443], pea and *Vigna* [438], lupin [444], and *Arabidopsis thaliana* [445]. For example, in *Lens culinaris*, Bermejo et al. [443] developed an efficient system to accelerate the number of annual generations. Embryos were taken at 15, 18, 21, and 24 DAP and grown on MS medium [72] with five different concentrations of BAP (0, 0.025, 0.05, 0.1, and 0.25 mg L^−1^). The results showed that 18 DAP was the most suitable time for embryo retrieval, given the high germination rates (13–70%). This high variability was correlated with genotype and culture medium. The absence of BAP in the culture medium was the most suitable for complete embryo maturation (41–87%). All the plants obtained were morphologically normal and fertile. Four generations per year were obtained with this approach.

In some cases, ER was used to facilitate the propagation of valuable ornamental plants. For example, although many *Agave* species produce seeds, it takes longer for the plants to reach appropriate maturity and size. Thus, two different kinds of explants were advantageous in initiating the multiplication process: zygotic embryos and small offshoots that grow around a mother plant [446].

In orchids, irregular germination in nature and in vitro is a major obstacle to the conservation of rare and endangered species [447]. Long-term storage of orchid seeds requires cryopreservation at very low temperatures, as their longevity under conventional storage conditions is not entirely reliable [448]. Kendon et al. [449] successfully collected mature, near-mature, and immature seeds of orchids (i.e., *Aerangis ellisii* and *Angraecum rutenbergianum*) from the Central Highlands of Madagascar. Seed capsules were collected in a sterile culture medium in the wild. In the laboratory, most seed capsules harvested by the in vitro collecting method yielded seeds in a good state of freshness that stayed sterile inside the capsules. Near-mature and immature seeds showed a high germination rate [449].

Mango (*Mangifera indica*) suffers from heavy fruit drop; therefore, embryo culture can improve breeding efforts [450]. Various factors are associated with fruit drop, including competition between developing fruits, poor nutrition, moisture stress, hormonal imbalance, and climatological factors, such as high temperature, rainfall during flowering, hailstorms, high wind speed, varietal factor, lack of fertilization, and biotic stresses [451]. In mangoes, the immature fruit technique (35–45 days old) for in vitro regeneration has improved reproduction efficiency [451]. Moreover, Chandra et al. [452] regenerated strongly immature mango embryos into complete plantlets using in vitro culture. Sahijram et al. [453] suggested that the collection of mango fruitlets at 42–56 DAP was the finest solution for embryo culture. In mango, the in vitro rescue of immature embryos can be also employed. In this case, the objectives concern the conservation of germplasm and the saving of hybrids produced by crosses to obtain traits of interest. Pérez-Hernández and Grajal-Martín [454] investigated the in vitro culture of embryos to rescue cultivars. The authors reported a success rate of 83% for the cultivars “Lippens” and “Keitt” after growing immature embryos in a liquid medium during the maturation period. Furthermore, it is necessary to highlight the interaction between the cultural system and the media composition used. More recently, Souza et al. [455] established an efficient protocol for in vitro rescue of immature embryos to preserve mango germplasm. The immature fruits were harvested at 13–20 DAP. After sterilization, embryos were extracted from the fruits and cultured in a half-strength MS salt medium [72] supplemented with 100 mg L^−1^ cysteine, 0.5 mg L^−1^ GA_3_, and 30 g L^−1^ sucrose. Cultures were maintained at 27 ± 1 °C, with a 16 h photoperiod. Seedlings were preserved in vitro or in vivo. For in vitro storage, viable seedlings were incubated at 22 ± 1 °C, with a 12 h photoperiod. The results demonstrated the feasibility of saving immature mango embryos and storing them in vitro for 12 months. For in vivo storage, after one year, the plants were removed from in vitro culture and transferred to a greenhouse in pots containing a commercial substrate. The substrate was saturated with a solution at three different indole-3-butyric acid (IBA) concentrations to induce better rooting. IBA did not affect the acclimatization of mango plants compared to plants raised without the hormone. In conclusion, a sufficient number of plants were obtained that developed normally.

*Lilium callosum* Sieb. et Zucc. is native to Taiwan and is a little-known species, as it has been considered extinct in the wild since 1915. At the beginning of the 20th century, only two specimens of *L. callosum* existed in Taiwan. However, the species was rediscovered in 2011 in eastern Taiwan. Therefore, a new interest has arisen to study and conserve this species. Once rediscovered, the species was micropropagated by culturing scales of the bulb on a modified basal MS medium [72] supplemented with 0.1 mg L^−1^ NAA, 0.1 mg L^−1^ BA, 1 g L^−1^ casein hydrolysate, and 3% sucrose. The explants were maintained at 25 ± 1 °C with a 12 h photoperiod and a light intensity of 5.6 μmol m^−2^ s^−1^ [456]. In addition, flowers from two regenerated plants were crossed by hand pollination, and immature ovules were cultured in vitro. In the plants obtained, the karyotype (2*n* = 2*x* = 24) was determined by the length of the chromosome arms, the position of the centromere, and the presence of satellite regions. *L. callosum* is a widespread species in Asia, distributed from Taiwan to northern Russia. However, phylogenetic analysis using ITS sequences demonstrated that *L. callosum* from Taiwan is not clustered with other *L. callosum* accessions [456]. A limiting effect on the fitness of *L. callosum* in Taiwan was found to be the inability to produce meagre numbers of seeds in the native habitat. This certainly limits the genetic variability in the population, leading to the phenomena of genetic drift. Chen et al. [456] used the ER strategy to conserve germplasm, and this approach could be critical for the conservation of *L. callosum* in Taiwan. The lily is a globally important commercial flower crop. Therefore, introducing new species in genetic improvement programs should broaden the genetic variability to select new varieties. In particular, the pure orange color of the flower of *L. callosum* is unique and could be useful to be introgressed into other species of the genus *Lilium*.

*Magnolia dealbata* is another unusual species considered extinct until 1977, when Vovides and Iglesias [457] rediscovered it. It is a species endemic to Mexico, but its distribution is restricted to tiny regions. Moreover, this species is included in the Red List [458]. Its seeds have low germinability and are viable for a limited time. In addition, this species has shown a reduced capacity for agamic regeneration [457]. Mata-Rosas et al. [459] developed a very efficient method of in vitro propagation from mature zygotic embryos. The best results were obtained by culturing mature embryos on WP medium [99] supplemented with 13.3 μM or 22 μM BA and 2.26 μM 2,4-D, while using 1 g L^−1^ PVP significantly reduced tissue necrosis. Somatic embryos were induced to direct and/or indirect regeneration in a high percentage (85% of cultured embryos). Several seedlings were acclimated in a greenhouse in pots. Plant survival was very high (90%). Therefore, it has been shown that in vitro culture could play a key role in the propagation and conservation of this endemic species.

## 5. Conclusions and Perspectives

The earliest attempts at in vitro plant tissue culture techniques now date back more than 120 years, but even today, the practical interest in the main objectives of crop breeding shows no signs of waning. In this context, ER strategies continue to be extremely attractive and the subject of numerous studies, as illustrated earlier in this review. This interest is undoubtedly justified by the many interesting applications of ER, as illustrated in Figure 1. As reported here, embryo culture has a central and strategic position in making basic and applied research feasible. Regarding the first type of applications, recall that in vitro recovery of lethal embryos has facilitated important investigations into the role of specific genes in plant physiology and development [460,461,462]. On the other hand, in terms of more directly oriented applications, it is worth noting ER’s role in transferring necessary genetic traits from wild species to crops [28]. Added to this is that applications of major biotechnologies have called for efficient in vitro regeneration techniques, a goal often achieved by developing in vitro culture systems with immature embryos, usually characterized by a greater capacity for adventitious morphogenesis [463,464,465,466]. In addition, in vitro culture of immature embryos has recently proven to be an effective option to recover improved plants through genome editing, currently one of the most attractive options in plant breeding (Figure 1) [467,468]. However, the scientific areas where ER techniques are applied could be further increased in the future as in the study of obligate parasitic plants with specific respect to the analysis of germination physiology and the description of embryo developmental stages. These species represent a very negative factor for the productivity of various crops. To date, the potential of in vitro tissue culture continues to be limited by the difficulty in obtaining results for certain plant genotypes recalcitrant to regeneration and genetic transformation. In this regard, the case of sunflower is emblematic [469]. The cell dedifferentiation is accompanied by changes at the chromatin level and the reprogramming of gene expression, highlighting the central role of epigenetic regulation in this process. However, the epigenetic regulatory pathway underlying cell differentiation/dedifferentiation still needs to be fully understood. Therefore, in some cases, it is plausible that the activation of the epigenetic mechanisms essential for a differentiated cell to regain cellular totipotency does not occur in the correct sequence [470]. In addition, finding the most appropriate growing conditions (temperature, photoperiod humidity, etc.) and the most effective chemical composition of culture media are practical issues that need to be resolved for different plant species [17,471,472]. Unfortunately, achieving real technical improvements in these areas, as well as selecting the most appropriate genotype, is often the result of tedious and sometimes empirical work. However, we believe that this area of research deserves to be re-evaluated and stimulated. Finally, speaking of the countless factors that influence ER performance, selecting the most suitable stages of embryonic development to ensure the recovery of viable seedlings plays a particularly important role in improving the efficiency of these techniques [136,472].

## Figures and Tables

**Figure 1 plants-12-03106-f001:**
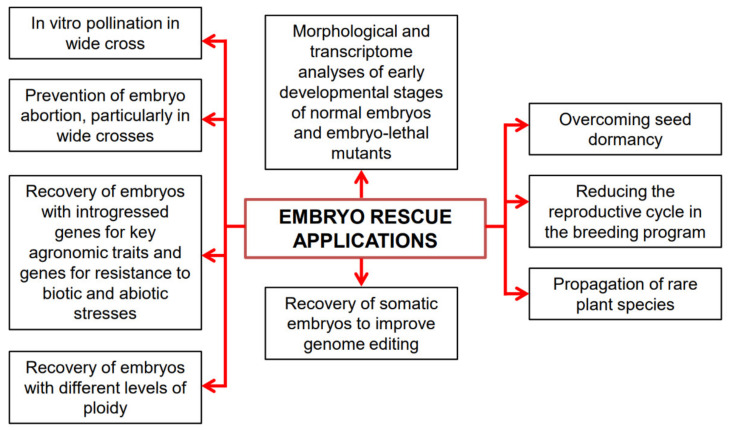
A diagrammatical representation of embryo rescue (ER) applications.

**Figure 2 plants-12-03106-f002:**
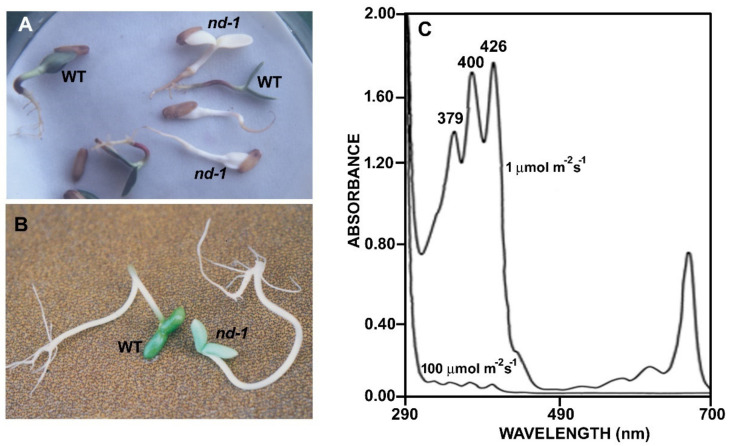
Germination of sunflower seeds under light intensity conditions of 100 µmol m^−2^ s^−1^ (**A**) or 1 µmol m^−2^ s^−1^ (**B**). (**C**) Absorption spectra of total pigments extracted from cotyledons of 10-day-old *nd-1* sunflower seedlings grown under high (100 µmol m^−2^ s^−1^) or very low (1 µmol m^−2^ s^−1^) light intensity conditions. (**C**) is modified from Conti et al. [423]. Adapted with permission (license number 5618090009667) from [423], obtained on 29 August 2023 from Oxford University Press. WT: wild type; *nd-1*: *non-dormant-1* mutant.

## Data Availability

Not applicable.

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
