# Peer review of "Embryo Rescue in Plant Breeding"

_plants, 2023, doi:10.3390/plants12173106_

Round 1

Reviewer 1 Report

The presentation in the article (plants-2439760) is routine type and seems like reviewed 10 years before. The article should be revised thoroughly so that it will create interest among the readers.

1.Abstract should be written again.

2. 1st paragraph under introduction requires improvement for introducing embryo rescue in the beginning sentences.

3.The review seems to be like 10 years before compiled and written.

4.No citation of recent works though several works on embryo rescue are reported in recent years.

5. The whole review presented in the article is just like a project report.

6. Please see the article. Many mistakes are observed in the article which are given as insert comments in the pdf text.

7.Whole article needs improvement on embryo rescue particularly using recent examples.

Minor editing

Author Response

Dear Reviewer,

I agree with you that the review looks like it was written 10 years ago. But you read and I am sorry about that the first submission and not the corrected version which is very updated in the citations.
I send you the new version that I had submitted on June 30. I am sorry for this inconvenience but this was not my fault.
I regret the inconvenience caused.
Claudio Pugliesi

DAFE, University of Pisa Italy

Reviewer 2 Report

This manuscript bring together research in ER. The manuscript have the merit of pulling to gether a considerable amount of published work in this area to a single review. However the manuscript is way too long and covers a huge area when talking about Er. Rather be published as a book chapter.

It would be better to have sub headings for Plant genus basis/or family to present the work. The deatils of all culture conditions etc are not highly necessary in the paragraphs, better summerised in tables.

In some sections the information is too detail, thus becomes very monotonus to read. 

For e.g section 3. Most of the paragraphs cn be summerised to tables, to make it easy for the reader. 

E.g. Applications of embryorescue; should be summerised.

The manuscript is well written. 

Author Response

ID: plants-2439760

Replay to Reviewer2

We sincerely thank the reviewer for constructive criticisms and valuable comments, which were of great help in revising the manuscript. Accordingly, the revised manuscript has been improved, where possible, with information required.

We send the revised manuscript. In the manuscript, corrections made are highlighted in yellow and when appropriate crossed out. English corrections or minor changes in the text are not reported so as not to make the text unintelligible.

First of all, it is important to note that the version of the manuscript analyzed by the reviewers does not correspond to the version subsequently sent to Plants in late June, following numerous detailed requests made by the Guest Editor of the Special Issue. The structure of the manuscript was subsequently modified extensively. Now, let's address the points raised by Reviewer2.

-) “This manuscript bring together research in ER. The manuscript have the merit of pulling together a considerable amount of published work in this area to a single review. However the manuscript is way too long and covers a huge area when talking about Er. Rather be published as a book chapter”

There are no doubts regarding the considerable length of our manuscript. However, the topic of Embryo Rescue (ER) is indeed vast, and in response to the Editor's requests, we developed a comprehensive content in the present version of manuscript. Specifically, the Editor asked for more technical details (e.g., the number of obtained plants) related to ER applied to major crops. In addition, the Guest Editor explicitly requested a description of methods for the recovery of induced haploids in cereals and an update of the literature in important crops (e.g., Brassicaceae family), which involved analysis of a significant number of scientific articles on the topics. Therefore, even after incorporating the Editor's suggestions, we believe that potential readers of Plants will find comprehensive tools to delve into the complex topic of ER in its entirety. Furthermore, ER is one of the earliest in vitro culture methodologies applied to plants, and it is still widely used in many genetic improvement programs, including modern ones that involve genome editing. In addition to this, we found it beneficial to include general information on embryogenesis in both monocots and dicots, the origin of the methodology, and the ER techniques applied to overcome post-zygotic barriers. Moreover, to make the vast topic less tedious and to summarize the main examples discussed, we included four tables showcasing the main applications of ER in the genetic improvement of monocots and dicots. In this context, we also provided a graphical representation of different ER-related applications in Figure 1.

-) “It would be better to have sub headings for Plant genus basis/or family to present the work.”

In line with the suggestions, we have now added several new sub-sections in Chapter 4 to channel various information and discussions on dicots and monocots. This choice allowed us to align with the divisions proposed in the four tables. The decision to prefer this basis of division over others, such as genus or botanical family, was made to avoid excessively altering the text while maintaining a distinction proposed by the Editor. However, the inclusion of these five new sub-sections necessitated a considerable revision of Chapter 4's structure, resulting in a renumbering of the citations.

-) “The deatils of all culture conditions etc are not highly necessary in the paragraphs, better summerised in tables. In some sections the information is too detail, thus becomes very monotonus to read.”

Regarding the possibility of tabulating information related to cultural conditions, we considered it but ultimately found it impractical and not advantageous for the reader. Therefore, we confirmed the inclusion of this type of information in the manuscript text (in paragraph 3.3 and other different sections).

-) “For e.g section 3. Most of the paragraphs can be summerised to tables, to make it easy for the reader. E.g. Applications of embryorescue; should be summerised.”

As for the current version of the manuscript, the structure of Chapter 3 is as follows: 3.1. Pre-zygotic Barriers in Hybridization and Techniques to Overcome Them; 3.2. Post-zygotic Barriers in Hybridization and Techniques to Overcome Them; 3.3. Medium Composition and Environmental Factors Suitable for Embryo Rescue Techniques. Regarding the opportunity to tabulate sub-section 3.3, we addressed this previously, and regarding 3.1 and 3.2 sub-sections, we believe that specific tables were not suitable. It is essential to emphasize that the applications of ER techniques are included in the current version of the manuscript in Chapter 4, along with the four tables provided, as well as the graphical representation in Figure 1.

Round 2

Reviewer 1 Report

Comments on MS ID: plants-2439760

There is substantial improvement in the review on embryo rescue. But, improvement in usage of English language is required as a lot of mistakes still exist in many sentences in the article. The intended meanings of those sentences are confusing. Please check the language of the article by a native English speaker.

Comments on MS ID: plants-2439760

There is substantial improvement in the review on embryo rescue. But, improvement in usage of English language is required as a lot of mistakes still exist in many sentences in the article. The intended meanings of those sentences are confusing. Please check the language of the article by a native English speaker.

Author Response

“There is substantial improvement in the review on embryo rescue. But, improvement in usage of English language is required as a lot of mistakes still exist in many sentences in the article. The intended meanings of those sentences are confusing. Please check the language of the article by a native English speaker.”

We sincerely thank the reviewer for constructive criticisms and valuable comments, which were of great help in revising the manuscript. Accordingly, the revised manuscript has been improved, with information required.

Since in this second version of our manuscript the main problems found by the reviewer were mainly grammatical and linguistic errors, the revision was done by a native English speaker. We tried to make some sentences that were difficult to understand clearer. All (major) changes are highlighted in light blue in the text. A thorough revision was also made to the bibliographical references. Again, changes are highlighted in light blue.

Round 3

Reviewer 1 Report

The review aricle ‘Embryo Rescue in Plant Breeding’ is ok in the current version and may be accepted for publication. However, the following points may be checked.

1.Line 259, proofread for brackets.

2. Line 279,280, 282 and other places, please change from ml L-1 to ml L-1

3.Line 290, 327, 329, 333, 352, 570, 436, 847 and other places for : change 11.5% to 11.5 % (Put gap before %)